# Hallucination-aware Intermediate Representation Edit in Large Vision-Language Models

**Wei Suo, Hanzu Zhang, Lijun Zhang, Ji Ma, Peng Wang**\* **& Yanning Zhang**
School of Computer Science, Northwestern Polytechnical University, China.
`{suowei,peng.wang,ynzhang}@nwpu.edu.cn`
`{jojo688,lijunzhang,maji}@mail.nwpu.edu.cn`

## Abstract

Large Vision-Language Models have demonstrated exceptional performance in multimodal reasoning and complex scene understanding. However, these models still face significant hallucination issues, where outputs contradict visual facts. Recent research on hallucination mitigation has focused on retraining methods and Contrastive Decoding (CD) methods. While both methods perform well, retraining methods require substantial training resources, and CD methods introduce dual inference overhead. These factors hinder their practical applicability. To address the above issue, we propose a framework for dynamically detecting hallucination representations and performing hallucination-eliminating edits on these representations. With minimal additional computational cost, we achieve state-of-the-art performance on existing benchmarks. Extensive experiments demonstrate the effectiveness of our approach, highlighting its efficient and robust hallucination elimination capability and its powerful controllability over hallucinations. Code is available at https://github.com/ASGO-MM/HIRE.

## 1 Introduction

Large Vision-Language Models (LVLMs) (Bai et al., 2023b; Dai et al., 2023; Liu et al., 2024b; Zhu et al., 2023) have made remarkable advancements in recent years, demonstrating the ability to generate context-aware language outputs based on visual understanding. By integrating visual information into Large Language Models (LLMs), LVLMs have demonstrated strong capabilities in tasks such as multimodal reasoning (Lu et al., 2022) and complex scene understanding (Luo et al., 2024). However, LVLMs currently face the issue of hallucination, where the generated responses contradict the actual visual content.

Recently, many approaches (Leng et al., 2024; Liu et al., 2024a; Yin et al., 2024b; Zhao et al., 2023; Ma et al., 2026) have been proposed to mitigate hallucinations in LVLMs, which can be broadly categorized into retraining methods and Contrastive Decoding (CD) methods. As illustrated in Fig. 1 (a), retraining methods aim to alleviate hallucinations by constructing specialized datasets that target hallucination phenomena and introducing new training paradigms (Jiang et al., 2024a; Fu et al., 2024; Lu et al., 2023). In contrast, Fig. 1 (b) illustrates the paradigm of CD methods, which mitigate hallucination during inference. The method can work by contrasting the output token probabilities of the original response with those of a weakened variant that is more susceptible to hallucination. CD methods effectively reduce hallucinations without requiring retraining or additional data (Leng et al., 2024; Huo et al., 2024; Manevich & Tsarfaty, 2024).

Although both methods have shown promising results in hallucination mitigation, they still face some notable limitations: 1) Retraining methods require substantial data collection and computational resources for partial or full model fine-tuning. However, both the high cost of dataset construction and the heavy computational burden (*i.e.,* they need to retrain the LVLM) limit the practical applicability of these methods. 2) Contrastive decoding methods, while more efficient in design, introduce high computational overhead. They require two forward passes per inference—one for generating the original output and another for producing a deliberately weakened variant prone to

---

\*Corresponding authors.

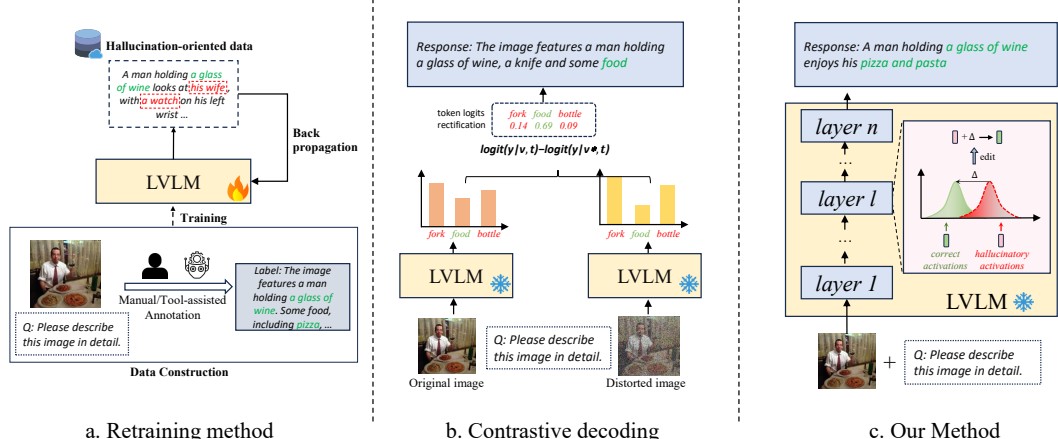

Figure 1: Comparison of mainstream hallucination mitigation paradigms. (a) Retraining-based methods: Constructing hallucination-specific datasets and training frameworks. (b) Contrastive decoding methods: Comparing the original probability distribution with a perturbed one. (c) Our method: Editing intermediate representations of LVLMs.

hallucination—which significantly increases latency. Moreover, they uniformly adjust the probability of all tokens without distinguishing whether a token is actually prone to hallucination. For instance, common tokens like "in," "on," or "from" rarely cause such issues. The one-size-fits-all adjustment leads to unnecessary computation and may harm output coherence. 3) Finally, hallucinations are not inherently harmful in tasks like creative writing (Jiang et al., 2024b), where appropriate hallucination can enhance expressiveness. However, most existing methods lack the ability to control the degree of hallucination, limiting their applicability in broader use cases where flexible generation is desired.

Recent studies on LLMs (Azaria & Mitchell, 2023; Chen et al., 2024b) have shown that a model's internal representations encode cues about the truthfulness of statements, enabling hallucination detection without external knowledge. Building on these insights, recent research on LVLMs (Li et al., 2024; Duan et al., 2025) demonstrates that similar techniques can be applied to multimodal settings, where hallucination detection can be effectively performed through simple classifiers trained on intermediate features. This suggests that authentic and hallucinatory features in LVLMs are clearly separable in the latent space. This raises a natural question: *Can we leverage this inherent separation to effectively eliminate hallucinations?*

Based on the above discussion, we propose **HIRE** (**H**allucination-aware **I**ntermediate **R**epresentation **E**dit), a feature-editing framework that dynamically detects and mitigates hallucinations at the intermediate representation level. As shown in Fig. 1 (c), instead of costly retraining, we propose to mitigate hallucinations at the feature level without modifying the LVLM's weights. Specifically, we first introduce a module, *Editor*, that learns to identify and isolate hallucination-related components by modeling both semantic invariance and hallucinatory differences between authentic and hallucinated responses, while preserving the underlying semantic information. Moreover, since uniformly editing all tokens leads to unnecessary computational overhead, we design a lightweight *Router* to selectively edit only tokens with high hallucination risk. Lastly, to enable more flexible hallucination control, we introduce a *Hallucination Regulator*, which allows dynamic control over hallucination levels via a simple hyperparameter.

Overall, our main contributions are: (1) We introduce a new paradigm for hallucination mitigation, which reduces fabricated content by modifying representations without retraining models or doubling inference costs. (2) We propose a new framework, HIRE, that dynamically detects and edits intermediate representations with high hallucination. Meanwhile, our method can control the degree of hallucinations to adapt to different user requirements by adjusting the editing intensity. (3) We validate the effectiveness of the proposed method through extensive experiments and achieve state-of-the-art performance on three benchmarks.

## 2 RELATED WORK

### 2.1 LARGE VISION-LANGUAGE MODELS

Building on the success of Large Language Models (LLMs) (Bai et al., 2023a; Brown et al., 2020; Chiang et al., 2023; Touvron et al., 2023) and cross-modal learning (Radford et al., 2021; Dosovitskiy et al., 2020), Large Vision-Language Models (LVLMs) achieve breakthrough performance by integrating visual perception and language generation capabilities, excelling in tasks such as image captioning (Hossain et al., 2019), visual question answering (Antol et al., 2015), and multimodal reasoning (Lu et al., 2022). A typical LVLM architecture comprises three core components: a visual encoder for hierarchically image feature extraction like CLIP (Radford et al., 2021); a cross-modal alignment module implemented via linear projection layers or advanced architecture like Q-Former (Li et al., 2023b); a large language model fine-tuned through instruction tuning for context-aware text generation. Despite these advancements, LVLMs inherently suffer from hallucination, where the generated text exhibits semantic inconsistencies with the input visual content.

### 2.2 MITIGATING HALLUCINATIONS IN LVLMS

Recently, multiple strategies have been proposed to address hallucination in LVLMs, which can be broadly categorized into retraining methods and Contrastive Decoding (CD) methods. Retraining methods retrain LVLMs using carefully constructed datasets and training paradigms specifically designed to address hallucinations. HA-DPO (Zhao et al., 2023) fine-tunes LVLMs using style-consistent hallucination sample pairs to promote non-hallucinatory outputs. HDPO (Fu et al., 2024) further targets diverse causes of hallucinations by constructing specialized preference pairs for visual distraction, long-context generation, and multimodal conflicts. In contrast, CD methods mitigate hallucinations by comparing the output distributions from the original and perturbed inputs, without requiring any model parameter updates. VCD (Leng et al., 2024) introduces noise into visual inputs to amplify hallucinations and suppresses them by contrasting perturbed and original token distributions. SID (Huo et al., 2024) retains only the least important visual tokens after shallow processing to amplify vision-text association hallucinations. Unlike previous works, we propose to mitigate hallucinations by detecting and editing hallucinated representations, avoiding the need for heavy training resources and the cost of dual inference.

## 3 PRELIMINARY

LVLMs have garnered significant attention due to their capacity to combine visual and textual information for generation tasks. These models take both an image and a textual prompt as input. Given an image, it is firstly processed by a visual encoder and a cross-modal interface. Subsequently, the visual information $v$ and the text-based query $q$ are combined and fed into the LLM for further processing. The LLM $p_\theta$ is a parameterized model that maps the input to a probability distribution over the next token. Generally, the LLM architecture typically consists of stacked Transformer layers (Vaswani et al., 2017), each comprising multi-head self-attention and a feed-forward network (FFN). After passing through all $L$ layers, the final hidden states are projected into the vocabulary space to produce a probability distribution $y_t$ over the next token. The above process can be formulated as:

$$y_t \sim p_\theta(y_t|v, q, y_{<t}), \tag{1}$$

where $y_t$ is the $t$-th generated token, and $y_{<t}$ represents the previous tokens. However, the generated sentences often suffer from hallucinations, where the output contradicts the real visual input.

## 4 METHOD

Current methods primarily employ retraining methods (Fu et al., 2024; Zhao et al., 2023) or CD methods (Leng et al., 2024; Wang et al., 2024) to mitigate hallucinations. However, retraining methods suffer from training resource burdens, while CD methods incur dual computational cost during inference. Moreover, these approaches are incapable of achieving controllable generation of hallucinations. Motivated by the separability of hallucinated and truthful representations in the latent

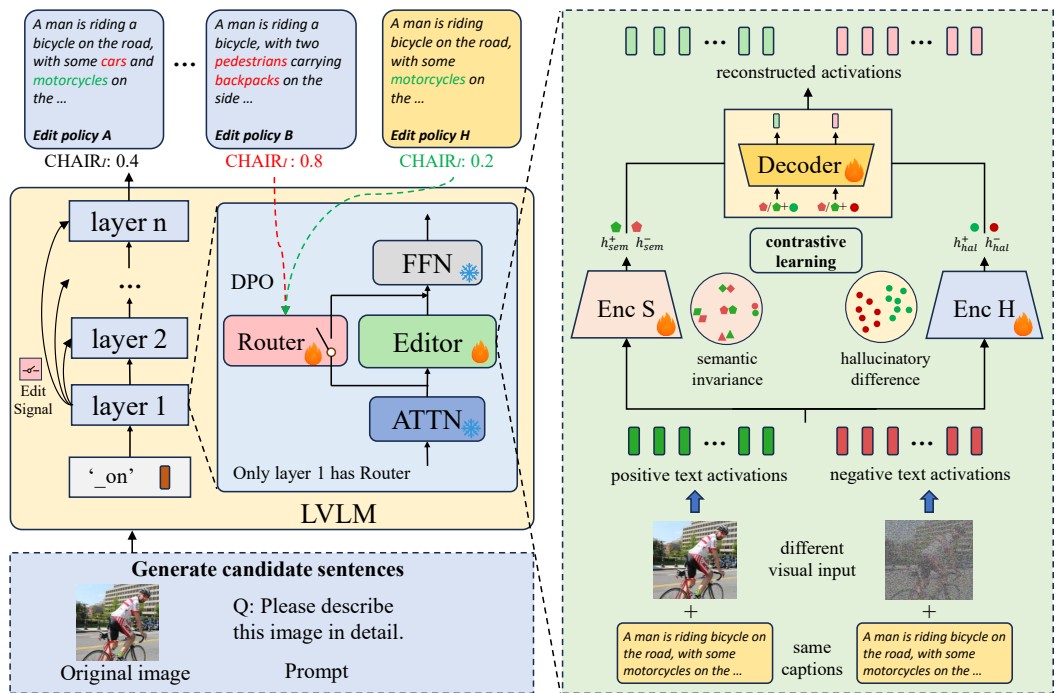

Figure 2: Overview of HIRE. Our framework consists of two key components: the Editor, which learns semantic invariance and hallucinatory difference through contrastive learning, and the Router, which learns efficient editing strategies through DPO.

space (Li et al., 2024; Duan et al., 2025), we propose a feature editing approach **HIRE**. As shown in Fig.2, HIRE dynamically identifies hallucination-prone components within representations and edits them by projecting features onto low-hallucination directions in the representation space.

To eliminate hallucination through editing and identify optimal editing strategies, we must address two key challenges: (1) how to determine the direction for obtaining hallucination-reduced representations, and (2) how to identify which token representations require editing. For the first challenge, we propose constructing hallucination-reduced representations by analyzing the divergence between high-hallucination and low-hallucination representations and then editing along this divergence direction. For the second challenge, our method autonomously learns effective editing strategies, enhancing the success rate of edits while minimizing ineffective changes. The following sections will detail our approach in terms of model architecture and optimization.

## 4.1 MODEL STRUCTURE

**Editor.** Based on the established correlation between hallucination phenomena and attention distributions (An et al., 2024; Huo et al., 2024), we identify the attention-layer representations at each transformer layer as our editing targets. The further analysis about the effect of editing different layers can be found in Appendix D.1. However, due to the entanglement between the representations of hallucinated and non-hallucinated texts (Jiang et al., 2024a), direct manipulation may disrupt semantic integrity(Li et al., 2023c; Chen et al., 2024c; Zhang et al., 2024). Inspired by (Zhang et al., 2024), we utilize a dual-encoder autoencoder $G_\phi$ that enables hallucination suppression while preserving semantics. Specifically, the $G_\phi$ consists of a semantic encoder $E_{\text{sem}}$, a hallucinatory encoder $E_{\text{hal}}$, a multi-head attention module $f_{\text{attn}}$, and a decoder $D$. Given the $t$-th token in attention-layer representations from $l$-th layer $h_{tl}$, the input representations are processed by both encoders to yield semantic representations $h_{tl,\text{sem}}$ and hallucinatory representations $h_{tl,\text{hal}}$, which can be formulated as:

$$h_{tl,\text{sem}} = E_{\text{sem}}(h_{tl}), \qquad h_{tl,\text{hal}} = E_{\text{hal}}(h_{tl}). \qquad (2)$$

To ensure correct editing, we compute the hallucination-reduction direction $\delta_l$ by averaging token-level differences between authentic and hallucinated representations within the hallucinatory subspace. The semantic and hallucinatory representations are fused and decoded to yield an optimized editing direction $\Delta_{tl}$, which is the direction of targeted feature editing. The $\Delta_{tl}$ are obtained as follows:

$$\Delta_{tl} = D\left(h_{tl,\text{sem}} + f_{\text{attn}}\left(h_{tl,\text{sem}}, h_{tl,\text{hal}} + \delta_l\right)\right) - D\left(h_{tl,\text{sem}} + f_{\text{attn}}\left(h_{tl,\text{sem}}, h_{tl,\text{hal}} - \delta_l\right)\right), \quad (3)$$

where the $f_{\text{attn}}$ denotes an attention fusion of $h_{tl,\text{sem}}$ (serving as query) and $h_{tl,\text{hal}}$ (serving as key and value). Overall, the $\delta_l$ aims to identify hallucination-reduced directions that are irrelevant to specific tokens within the hallucinatory subspace, while the $\Delta_{tl}$ leverages the $\delta_l$ to obtain token-specific hallucination-reduced directions in the original representation space.

**Router.** To avoid unnecessary editing operations on hallucination-free representations, we introduce a lightweight module, Router $R_\theta$. Motivated by the observed layer redundancy in LVLMs, where deeper layers tend to exhibit high similarity (Wu et al., 2023; Suo et al., 2024), as well as by recent findings indicating that shallow layers preserve more informative representations (Wang et al., 2025; Chen et al., 2024a), we employ an MLP that takes the first-layer transformer representation $h_{t0}$ as input and produces a binary decision $c$. This signal determines whether editing is required: if $c = 1$, the Editor is activated for all subsequent layers; otherwise, no editing is performed. Further experiments exploring alternative layer-wise decision strategies are provided in Appendix D.2]. With an editing strength $\alpha$ introduced, the editing process can be formulated as follows:

$$h_{tl,\text{aug}} = \begin{cases} h_{tl} + \alpha \cdot \Delta_{tl} & \text{if } c = 1, \\ h_{tl} & \text{if } c = 0, \end{cases} \quad (4)$$

where $h_{tl,\text{aug}}$ denotes edited representations, and the $\alpha \in [-1, 1]$ controls the direction and intensity of editing. A positive $\alpha$ suppresses hallucination by steering features toward low-hallucination directions, while a negative $\alpha$ amplifies them, enabling flexible control over hallucination levels.

## 4.2 MODEL OPTIMIZATION

The above computation reveals that the Editor $G_\phi$ and the Router $R_\theta$ are critical to detecting and correcting hallucination-related features. However, optimizing them is challenging due to the lack of labeled training data. Therefore, we introduce contrastive learning and DPO strategy to train the Editor $G_\phi$ and the Router $R_\theta$, respectively. To disentangle hallucinatory patterns from semantic content, we use contrastive learning to optimize the Editor $G_\phi$. Specifically, $H_{l,\text{sem}} = \{h_{tl,\text{sem}}\}_{t=1}^T$ ($T$ denotes the number of tokens) should have high similarity to semantically identical representations and low similarity to those with different meanings. Conversely, $H_{l,\text{hal}} = \{h_{tl,\text{hal}}\}_{t=1}^T$ exhibits high similarity within group similarity (hallucinated or authentic), but low cross-group similarity regardless of token semantics. To optimize the Router $R_\theta$, we maximize the likelihood ratio between effective edits and ineffective ones by introducing DPO (Rafailov et al., 2023), which learns from pairwise data by increasing the likelihood of preferred responses while suppressing less desirable ones.

### 4.2.1 DATA CONSTRUCTION

Based on the finding that visual uncertainty amplifies hallucinations in LVLMs (Leng et al., 2024), we obtain authentic and hallucinated representations by pairing identical textual inputs with both intact and visually-degraded images. The positive sample uses the original image to produce more faithful outputs, while the negative sample uses a noised version of the image, weakening visual grounding and increasing hallucination. The resulting intermediate representations are denoted as $H_l^+ = \{h_{tl}^+\}_{t=1}^T$ and $H_l^- = \{h_{tl}^-\}_{t=1}^T$. Our framework is also compatible with other hallucination induction techniques; comparative experiments are provided in the Appendix D.3.

To obtain pairwise preference data, we generate $N$ candidate captions for each image by applying different editing action sequences during greedy decoding, where each sentence $S_n$ of length $T_n$ has corresponding representations $\{h_{0t}\}_{t=1}^{T_n}$ and editing decision trajectory $\{c_t\}_{t=1}^{T_n}$. We quantify hallucination severity using the CHAIR$_I$ metric (Rohrbach et al., 2018). The most and least faithful sentences are used to construct preference pairs $(h^+, c^+)$ and $(h^-, c^-)$ for the Router training.

### 4.3 TRAINING PROCESS

**Editor.** We apply contrastive learning to both the semantic and hallucinatory encoders. The semantic encoder aims to preserve semantic information by maximizing the similarity between corresponding tokens in positive and negative samples, while minimizing the similarity for non-matching tokens. In contrast, the hallucinatory encoder focuses solely on hallucination-related signals and is trained to separate positive and negative samples regardless of their semantic similarity. The loss formulations are defined as follows, where negative samples are formulated similarly:

$$
\begin{aligned}
L^+_{tl,\text{sem}} &= \mathcal{L}_{\text{InfoNCE}}\left(h^+_{tl,\text{sem}},\ h^-_{tl,\text{sem}},\ H^+_{l,\text{sem}}\right), \\
L^+_{tl,\text{hal}} &= \mathcal{L}_{\text{InfoNCE}}\left(h^+_{tl,\text{hal}},\ H^+_{l,\text{hal}},\ H^-_{l,\text{hal}}\right),
\end{aligned}
\tag{5}
$$

where $\mathcal{L}_{\text{InfoNCE}}$ (Oord et al., 2018) encourages the anchor to be closer to positive samples than to negatives, $h^\pm_{tl,\text{sem}}$ and $h^\pm_{tl,\text{hal}}$ denote the semantic and hallucinatory embeddings of token $t$ at layer $l$, and $H^\pm_{l,\text{sem}}$, $H^\pm_{l,\text{hal}}$ are the sets of token embeddings $h^\pm_{tl,\text{sem}}$ and $h^\pm_{tl,\text{hal}}$. The learned semantic and hallucinatory representations are fused through a multi-head attention $f_{\text{attn}}$ and decoded to predict the original feature. To ensure both faithful reconstruction and effective editing, a reconstruction loss and an editing loss can be formulated as follows, where negative samples are calculated similarly:

$$
\begin{aligned}
L^+_{tl,\text{recon}} &= \text{MSE}(h^+_{tl}, D(h^+_{tl,\text{sem}} + f_{\text{attn}}(h^+_{tl,\text{sem}}, h^+_{tl,\text{hal}}))), \\
L^+_{tl,\text{edit}} &= \text{MSE}(h^+_{tl}, D(h^-_{tl,\text{sem}} + f_{\text{attn}}(h^-_{tl,\text{sem}}, h^+_{tl,\text{hal}}))),
\end{aligned}
\tag{6}
$$

where $h^\pm_{tl}$ denote the original representations of the token $t$ at layer $l$ of positive and negative samples. Averaging the sum of four loss terms across tokens and layers yields the final training objective $\mathcal{L}_e$.

**Router.** To guide the model towards optimal editing strategies, we adopt Direct Preference Optimization (DPO) (Rafailov et al., 2023), which reformulates reward maximization as a policy likelihood ranking problem. Unlike standard DPO that requires pairwise comparisons between a learnable policy $\pi_\theta$ and a reference model $\pi_{\text{ref}}$, our implementation eliminates the reference model based on recent findings (Meng et al., 2024) demonstrating its removability. The simplification is suitable for training the Router from scratch rather than fine-tuning an existing policy. Given paired state-action sequences $(h^+, c^+)$ (preferred) and $(h^-, c^-)$ (non-preferred), the optimization objective becomes:

$$
\mathcal{L}_{\text{r}} = -\mathbb{E}_{(h,c)}\left[\log \sigma\left(\beta\left(\log \pi_\theta(h^+, c^+) - \log \pi_\theta(h^-, c^-)\right)\right)\right],
\tag{7}
$$

where $\mathcal{L}_{\text{r}}$ denotes the Router's training loss, $\pi_\theta$ represents the learnable policy (Router), $\sigma(\cdot)$ is the sigmoid function, and $\beta = 0.1$ serves as a scaling factor regulating optimization intensity.

## 5 EXPERIMENTS

### 5.1 EXPERIMENTAL SETTINGS

**Dataset.** We evaluate our method on three benchmarks: CHAIR (Rohrbach et al., 2018), POPE (Li et al., 2023d), and AMBER (Wang et al., 2023). CHAIR measures object hallucination in image captions, calculating the proportion of objects mentioned in the text but missing from the ground-truth annotations. It provides object-level (CHAIR*I*) and sentence-level (CHAIR*S*) scores, with lower values indicating fewer hallucinations. We evaluate on 500 images randomly sampled from the MSCOCO (Lin et al., 2014) dataset. POPE assesses object hallucination in multimodal question answering using three object-sampling strategies (random, popular, adversarial) and binary queries like "Is there a <object> in the image?". Evaluation is based on 9,000 question-answer pairs from MSCOCO. AMBER evaluates hallucinations in LVLMs, using 1,004 generative instances and 14,216 discriminative instances annotated for existence, attribute, and relation hallucinations, providing a comprehensive framework for hallucination identification and classification.

**Implementation Details**. We construct the training samples for both the Editor and the Router using the training split of the MSCOCO dataset (Lin et al., 2014). Specifically, for the Editor, training is

Table 1: Evaluation results on the CHAIR benchmark with LLaVA-1.5.

| Method | LLaVA-1.5(Liu et al., 2024b) | | | InstructBLIP(Dai et al., 2023) | | |
|---|---|---|---|---|---|---|
| | $\text{CHAIR}_S \downarrow$ | $\text{CHAIR}_I \downarrow$ | TFLOPs↓ | $\text{CHAIR}_S \downarrow$ | $\text{CHAIR}_I \downarrow$ | TFLOPs↓ |
| baseline | 51.3 | 16.8 | 10.23 | 51.0 | 24.2 | 2.77 |
| *Max new tokens set to 512* | | | | | | |
| OPERA(Huang et al., 2024) | 45.2 | 12.7 | - | 47.4 | 12.9 | - |
| SID(Huo et al., 2024) | 44.2 | 12.2 | - | 42.3 | 12.4 | - |
| ICD(Wang et al., 2024) | 47.4 | 13.9 | 20.63 | 46.3 | 15.3 | 5.64 |
| VCD(Leng et al., 2024) | 46.8 | 13.2 | 20.46 | 44.0 | 13.6 | 5.61 |
| M3ID(Favero et al., 2024) | 48.3 | 13.5 | 20.71 | 45.2 | 13.9 | 5.70 |
| AVISC(Huo et al., 2024) | 45.2 | 13.4 | 20.08 | 43.9 | 13.5 | 5.63 |
| Octopus(Suo et al., 2025) | 39.2 | 11.1 | 21.39 | 40.6 | 12.1 | 5.87 |
| Projectaway(Jiang et al.) | 42.0 | 12.2 | - | 43.8 | 12.5 | - |
| VTI(Liu et al., 2025) | 35.8 | 11.1 | - | 43.4 | 11.8 | - |
| Ours(HIRE) | **30.2** | **9.7** | **11.81** | **39.0** | **11.5** | **3.45** |
| *Max new tokens set to 64* | | | | | | |
| baseline | 20.4 | 6.2 | 8.91 | 25.4 | 8.2 | 1.89 |
| M3ID+DPO(Favero et al., 2024) | **13.5** | 5.7 | - | - | - | - |
| Nullu(Yang et al., 2025) | 17.0 | 5.9 | - | - | - | - |
| Ours (HIRE) | 15.2 | **5.4** | 9.18 | **14.8** | **5.4** | 2.32 |

performed for 5 epochs using a subset of 2,000 samples. For the Router module, we train for 100 epochs using 8,000 samples. Both modules employ the SGD optimizer (Bottou, 2010) with an initial learning rate of $1 \times 10^{-2}$, followed by a Cosine Annealing scheduler reaching a minimum learning rate of $1 \times 10^{-3}$. For the Router, we specifically set the group size to 10. We report the results of our method with a consistent edit strength $\alpha$ set to 1. All experiments were performed on four 3090 GPUs. Further discussions on the different training configurations are provided in Appendix D.5.

## 5.2 RESULTS ON BENCHMARKS

We conduct experiments on LLaVA-1.5 (Liu et al., 2024b) and InstructBLIP (Dai et al., 2023) across three widely used benchmarks: CHAIR (Rohrbach et al., 2018), POPE (Li et al., 2023d), and AMBER (Wang et al., 2023), with related results from (Suo et al., 2025; Xing et al., 2024; Huo et al., 2024; Jiang et al., 2024a; Liu et al., 2025; Jiang et al.). Overall, our method consistently outperforms existing approaches while maintaining efficient inference. As shown in Table 1, it reduces sentence-level and instance-level hallucinations by ∼40% and ∼50% respectively on both LLaVA-1.5 and InstructBLIP in the long description scene (max new tokens set to 512). Additionally, in the short description scene (max new tokens set to 64), our method also demonstrates reliable hallucination mitigation capabilities. Table 2 shows that our method achieves improvements of 1.48/3.79 and 0.48/1.99 over the state-of-the-art method Octopus(Suo et al., 2025) on the two models on the accuracy and F1 score, demonstrating superior performance in discriminative tasks. As shown in Table 3, our method achieves the highest AMBER scores-an indicator that averages performance across generative and discriminative tasks. Compared to the baseline, it significantly improves AMBER scores by 7.54 on LLaVA-1.5 and 6.38 on InstructBLIP. In addition, benefiting from the lightweight nature of our editing strategy and the selective token editing enabled by the Router, our method introduces minimal inference overhead. In summary, our approach effectively mitigates hallucination in both generative and discriminative tasks, with only a small additional overhead. Additional analysis and results on generalization and stability can be found in Appendix D.6 and Appendix D.7.

## 5.3 HALLUCINATION REGULATOR

Recognizing that hallucinations can have positive effects in certain scenarios like creative writing (Jiang et al., 2024b), it is important to have a controllable generation of hallucinations. However, existing methods exhibit unstable hallucination control capabilities. Detailed experimental results are provided in Appendix D.8. Hence, we conduct experiments on the edit strength hyperparameter

Table 2: Evaluation results on the POPE benchmark on the MS COCO datasets with LLaVA-1.5 and InstructBLIP.

| Method | Random | | Popular | | Adversarial | | ALL | | TFLOPs↓ |
|---|---|---|---|---|---|---|---|---|---|
| | Acc. | F1 | Acc. | F1 | Acc. | F1 | Acc. | F1 | |
| LLaVA-1.5-7B | 83.77 | 81.94 | 82.57 | 80.86 | 79.77 | 78.47 | 82.04 | 80.42 | 8.13 |
| *Referenced Results* | | | | | | | | | |
| +HACL | 88.59 | 88.70 | 87.84 | 87.36 | 86.54 | 85.73 | 87.66 | 87.26 | - |
| +VTI | 89.50 | 88.89 | 87.36 | 86.69 | 82.57 | 82.11 | 86.48 | 85.90 | - |
| *Comparable Results* | | | | | | | | | |
| +ICD | 87.51 | 83.28 | 83.15 | 83.91 | 79.13 | 80.41 | 83.26 | 82.53 | - |
| +ConVis | 84.70 | - | 83.20 | - | 81.10 | - | 83.00 | - | - |
| +OPERA | 84.40 | - | 83.40 | - | 81.20 | - | 83.00 | - | - |
| +VCD | 85.43 | 83.99 | 83.17 | 81.94 | 80.27 | 79.49 | 82.96 | 81.81 | 16.26 |
| +M3ID | 86.13 | 81.85 | 82.07 | 80.77 | 79.50 | 78.15 | 82.57 | 80.26 | 16.26 |
| +AVISC | 84.67 | 82.21 | 83.67 | 81.27 | 81.83 | 79.55 | 83.39 | 81.01 | 16.26 |
| +Octopus | 87.51 | 85.40 | 85.20 | 84.19 | 82.22 | 81.44 | 85.79 | 83.44 | 16.34 |
| +Ours | **90.37** | **90.25** | **87.70** | **87.86** | **83.73** | **83.56** | **87.27** | **87.23** | **10.62** |
| InstructBLIP | 81.53 | 81.19 | 78.47 | 78.75 | 77.43 | 78.00 | 79.14 | 79.31 | 1.08 |
| +ICD | 84.36 | 83.82 | 77.88 | 78.70 | 75.17 | 77.23 | 79.14 | 79.92 | - |
| +OPERA | 84.57 | 83.74 | 78.24 | 79.15 | 74.59 | 76.33 | 79.13 | 79.74 | - |
| +VCD | 82.03 | 81.56 | 79.13 | 79.20 | 77.23 | 77.72 | 79.46 | 79.49 | 2.16 |
| +M3ID | 82.33 | 81.53 | 80.90 | 80.42 | 78.53 | 78.49 | 80.59 | 80.15 | 2.16 |
| +AVISC | 86.03 | 84.41 | 84.27 | 82.77 | 81.83 | 80.67 | 84.04 | 82.62 | 2.16 |
| +Octopus | 86.63 | 85.30 | **84.90** | 83.55 | **82.83** | 81.43 | 84.79 | 83.43 | 2.27 |
| +Ours | **90.30** | **89.83** | 84.03 | **84.25** | 81.47 | **82.17** | **85.27** | **85.42** | **1.25** |

Table 3: Evaluation results on the AMBER benchmark with LLaVA-1.5 and InstructBLIP.

| Model | Setting | Generative | | | | | Discriminative | | | AMBER↑ |
|---|---|---|---|---|---|---|---|---|---|---|
| | | CHAIR↓ | Cover↑ | Hal↓ | Cog↓ | TFLOPs↓ | Acc.↑ | F1↑ | TFLOPs↓ | |
| LLaVA-1.5 | baseline | 8.0 | 44.5 | 31.0 | 2.2 | 9.89 | 67.00 | 71.10 | 8.07 | 81.58 |
| | VCD | 6.7 | 46.5 | 27.8 | 2.0 | 19.55 | 67.30 | 71.10 | 16.14 | 82.20 |
| | M3ID | 6.0 | 48.9 | 26.0 | 1.5 | 19.81 | 67.25 | 70.90 | 16.14 | 82.25 |
| | AVISC | 6.3 | 46.6 | 25.6 | 2.0 | 19.45 | 70.70 | 75.45 | 16.14 | 84.60 |
| | Octopus | 4.8 | 49.2 | 23.4 | **1.2** | 20.48 | 76.70 | 82.70 | 16.35 | 88.95 |
| | Ours | **4.6** | **49.9** | **20.4** | 1.5 | **11.73** | **79.20** | **82.83** | **10.58** | **89.12** |
| InstructBLIP | baseline | 8.4 | 46.4 | 31.1 | 2.6 | 2.93 | 68.20 | 74.60 | 1.04 | 83.10 |
| | VCD | 7.6 | 47.7 | 29.9 | 2.2 | 6.11 | 69.65 | 75.90 | 2.08 | 84.15 |
| | M3ID | 6.9 | 47.2 | 27.5 | 2.2 | 5.87 | 69.05 | 75.25 | 2.08 | 84.20 |
| | AVISC | 6.7 | 46.7 | 28.0 | 2.6 | 6.09 | 72.60 | 78.60 | 2.08 | 85.95 |
| | Octopus | 6.1 | 48.5 | **22.2** | **1.3** | 6.47 | 74.00 | 79.70 | 2.19 | 86.80 |
| | Ours | **5.3** | **49.3** | 23.8 | 1.8 | **3.57** | **78.34** | **84.26** | **1.21** | **89.48** |

$\alpha$ in formulation 4 to investigate its critical role in hallucination control for LVLMs. As shown in Figure 3, we conduct experiments with different values of $\alpha$, ranging from -0.7 to 1.0. It can be found that a positive $\alpha$ value can mitigate hallucination at both sentence-level and object-level granularities, and the reduction of hallucination increases with higher values. Conversely, negative $\alpha$ values demonstrate a proportional amplification effect on hallucination rates, with lower values inducing stronger hallucinatory responses. From the figure, it can be observed that our method has excellent controllability in hallucination generation.

## 5.4 ABLATION STUDY

We conduct ablation studies on CHAIR using LLaVA-1.5 to validate each module's contribution. As shown in Table 4, the original model's results are provided first. Adding only the semantic encoder

Table 4: Ablation study on the impact of encoder architecture and the Router module. Results show the effects on hallucination mitigation performance and computational cost.

| | Semantic encoder | Hallucinatory encoder | Router | CHAIR$_S\downarrow$ | CHAIR$_I\downarrow$ | TFLOPs$\downarrow$ |
|---|---|---|---|---|---|---|
| 1 | | | | 51.3 | 16.8 | 10.23 |
| 2 | ✓ | | | 37.0 | 12.2 | - |
| 3 | | ✓ | | 48.6 | 13.0 | - |
| 4 | ✓ | ✓ | | 30.4 | **9.5** | 16.23 |
| 5 | ✓ | ✓ | ✓ | **30.2** | 9.7 | 11.81 |

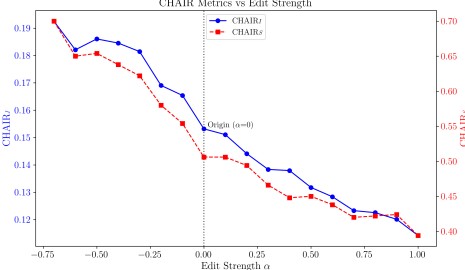

Figure 3: Control hallucination via $\alpha$.

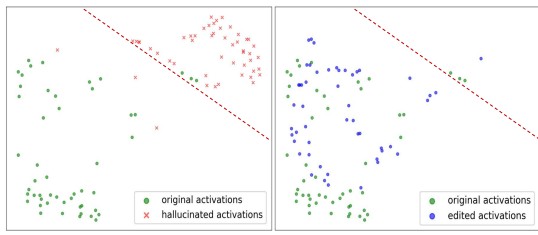

Figure 4: Distribution of original, hallucinated, and edited representations.

or hallucinatory encoder individually yields limited gains. Combining both achieves the lowest hallucination rate, demonstrating their complementary roles. Furthermore, integrating the Router reduces computational cost by ∼30% without performance loss, underscoring its efficiency. These results confirm the necessity of both encoders for effective hallucination mitigation and the Router's role in maintaining efficiency.

## 5.5 QUALITATIVE EVALUATION

To demonstrate our method's effectiveness in suppressing hallucination within the representation space, Figure 4 compares distributions of non-hallucinated (green), hallucinated (red), and edited (blue) feature. Non-hallucinated features are extracted using the image and its ground-truth caption. Following Wang et al. (2024), we introduce hallucination into features via a disturbed prompt and hallucinated caption to avoid image perturbation bias. Edited representations result from applying our Editor to hallucinated ones. The left subfigure shows clear separation between hallucinated and non-hallucinated clusters in the hallucinatory subspace. On the right, edited representations shift toward the non-hallucinated cluster and exhibit overlap and converge, confirming that our editing effectively reduces hallucination. Moreover, Fig. 5 presents representative examples from both generative tasks and discriminative tasks on the MSCOCO dataset. Hallucinated content in the figure is highlighted in red. It can be observed that our method eliminates hallucinations and provides a more accurate interpretation of the image. More results are provided in the Appendix E.

## 5.6 COMPATIBILITY WITH STEERING-BASED METHODS

Several steering-based methods exist for hallucination mitigation. For example, Nullu (Yang et al., 2025) operates by editing model weights and Projectaway (Jiang et al.) removes hallucinations from image features. In contrast, our method focuses on the textual feature space and further making fine-grained editing decisions per token. Hence, our approach can be integrated with other steering-based methods, offering a composite solution. Following the experimental settings of these methods, we evaluate the combined performance using the CHAIR metric across different max new token constraints. As shown in Table 5 and Table 6, our method can be combined with steering-based methods to yield stronger hallucination suppression.

## 5.7 EVALUATION ON GENERAL CAPABILITIES

We evaluate the general capabilities of the model equipped HIRE, using LLaVA-1.5-7B as a representative model on two challenging benchmarks: MME (Yin et al., 2024a) and SEED-Bench (Li

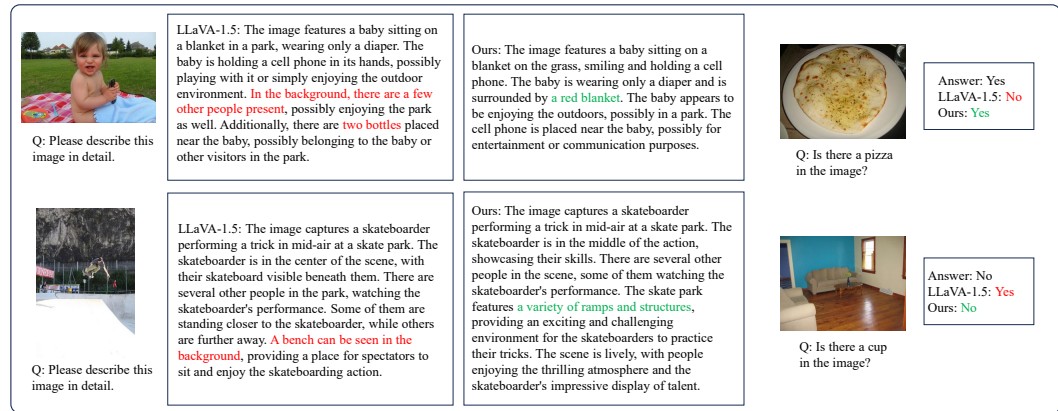

Figure 5: Some examples of generative and discriminative tasks on the MSCOCO dataset, with hallucinated content highlighted in red and newly added correct content displayed in green.

Table 5: Evaluation results on the CHAIR with Max New Tokens set to 512.

| Method | CHAIR$_S\downarrow$ | CHAIR$_I\downarrow$ |
|---|---|---|
| LLaVA-1.5-7B | 51.3 | 16.8 |
| +Projectaway | 43.8 | 12.5 |
| +Ours | 30.2 | 9.7 |
| +Ours & Projectaway | **27.6** | **8.3** |

Table 6: Evaluation results on the CHAIR with Max New Tokens set to 64.

| Method | CHAIR$_S\downarrow$ | CHAIR$_I\downarrow$ |
|---|---|---|
| LLaVA-1.5-7B | 20.4 | 6.2 |
| +Nullu | 17.0 | 5.9 |
| +Ours | 15.2 | 5.4 |
| +Ours & Nullu | **13.2** | **4.6** |

Table 7: Comparison of general capabilities on MME and SEED-Bench.

| Method | MME $\uparrow$ | SEED $\uparrow$ |
|---|---|---|
| LLaVA-1.5 7B | 1751.64 | 64.3 |
| +Ours | 1751.99 | 63.8 |

et al., 2023a).As shown in Table 7, it can be observed that our method preserves the model's general capabilities on these benchmarks.

# 6 CONCLUSION AND LIMITATIONS

In this paper, we propose a novel adaptive feature-editing framework that dynamically detects and calibrates hidden-layer activations to either suppress or enhance hallucinations, forming flexible, input-aware workflows. Our method operates in a single inference pass without updating parameters of LVLMs, offering exceptional efficiency while maintaining strong performance. It is also easily extendable to tasks requiring flexible control over hallucinations. We expect that this work will provide a general, practical paradigm for mitigating hallucination across diverse scenarios.

Despite these advantages, we note a limitation of the current framework. Our method employs all tokens and layers for training, a process that can be susceptible to noise. Training selectively on the most critical hidden states presents a significant opportunity for enhancing data efficiency. This work, as an initial step, aimed primarily at validating the feasibility of the proposed editing paradigm. We will explore the aforementioned direction in future work.

ACKNOWLEDGEMENT

This work is supported in part by the Natural Science Basic Research Program of Shaanxi Province (No. 2024JC-DXWT-07).

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

## A    DETAILED EXPERIMENTAL SETTINGS

### A.1    VISUAL PERTURBATION METHODS

We follow VCD (Leng et al., 2024) to apply visual perturbations by introducing noise to images. For LLaVA-1.5, we set the noise step to its maximum value of 999, which lies within its supported range of 0–999. For InstructBLIP, considering that the Q-former module relies on interactions between textual and visual inputs, excessively corrupted images can introduce unwanted degradation into the representations. To mitigate this, we limit the noise step to 600.

## B    LICENSE OF ASSETS

LLaVA-1.5 (Liu et al., 2024b) is available under the Apache-2.0 License and InstructBLIP (Dai et al., 2023) is available under BSD-3-Clause License. CHAIR is under the BSE 2-Clause License. POPE is available under the MIT License and AMBER is available under Apache-2.0 License.

## C    EVALUATION METRIC

**AMBER.** We follow the experimental setup in (Suo et al., 2025). In the generative task of AMBER, we report four metrics: CHAIR, Cover, Hal, and Cog.

*CHAIR* measures the proportion of objects mentioned in the generated sentences but not present in the ground-truth labels. Specifically, given the list of generated objects $G_{obj} = \{obj_1^G, obj_2^G, \ldots, obj_n^G\}$ and the list of annotated objects $A_{obj} = \{obj_1^A, obj_2^A, \ldots, obj_n^A\}$. *CHAIR* is calculated by the following formula:

$$CHAIR = 1 - \frac{len(G_{obj} \cap A_{obj})}{G_{obj}}. \tag{8}$$

*Cover* measures the ratio of between the correctly mentioned objects in responses and the total num of objects in the annotations:

$$Cover = \frac{len(G_{obj} \cap A_{obj})}{len(A_{obj})}. \tag{9}$$

*Hal* indicates whether a response contains hallucination. For each response, Hal is defined as 1 when the CHAIR score is greater than 0, and 0 otherwise, as shown below:

$$Hal = \mathbb{I}(CHAIR > 0), \tag{10}$$

**Cog** measures the alignment between model-generated hallucinations and those identified by human cognition. Specifically, AMBER defines a target set of hallucinated objects as $H_{obj} = \{obj_1^H, obj_2^H, \ldots, obj_m^H\}$. The Cog score is computed as the proportion of hallucinated objects generated by the model that also appear in the human-annotated set, as shown below:

$$Cog = \frac{|G_{obj} \cap H_{obj}|}{|G_{obj}|}. \tag{11}$$

## D    ADDITIONAL EXPERIMENTS

### D.1    EDIT LAYERS FOR HIRE'S EDITOR

To investigate the influence of different editing layer, Table 8 shows the hallucination mitigation performance with three different editing layer sets, where shallow layers represent the editing layers of 0∼9, middle layers represent the editing layers of 10∼19, deep layers represent the editing layers

Table 8: Evaluating the impact of editing different layers on the CHAIR benchmark.

| Edit layer | CHAIR$_S$ ↓ | CHAIR$_I$ ↓ |
|---|---|---|
| baseline | 51.3 | 16.8 |
| shallow (0∼9) | 42.0 | 13.0 |
| middle (10∼19) | 36.0 | 10.7 |
| deep (20∼29) | 46.8 | 14.1 |
| HIRE (0∼31) | **30.2** | **9.7** |

Table 9: Performance of different router strategies evaluated by hallucination (CHAIR) and text quality (BLEU).

| Router type | CHAIR$_S$ ↓ | CHAIR$_I$ ↓ | BLEU ↑ | training time ↓ |
|---|---|---|---|---|
| LLaVA-1.5 | 51.3 | 16.8 | 17.47 | - |
| I (a unified router) | 30.6 | 9.6 | 20.53 | 11h |
| II (layer-specific routers) | 30.4 | 9.6 | 20.65 | 14h |
| III (initial embedding-based decision) | 46.2 | 12.9 | 16.93 | 8h |
| Ours (first layer embedding-based decision) | 30.2 | 9.7 | 20.59 | 8h |

of 20∼29. The results indicate that editing the middle layers plays the most significant role in mitigating hallucinations, while editing shallow layers has a moderate effect. Editing deep layers, however, shows minimal contribution to performance gains. However, editing all layers achieves superior performance compared to editing only partial layers.

## D.2 EXPERIMENTS ON LAYER-WISE CONTROL

We conduct comparative studies about router decision strategies, comparing with: (I) **a unified router** shared across all tokens and layers; (II) **layer-specific routers**: each layer owns an independent router; (III) **initial embedding-based decision**: the router determines editing decisions for all subsequent layers based solely on the input embedding of the LVLM. To evaluate the quality of responses generated by our method, we further incorporate the BLEU metric (Papineni et al., 2002) (where higher scores indicate better sentence quality) to assess semantic integrity and coherence. The corresponding results are summarized in Table 9. We observe that both Strategy I and Strategy II achieve performance comparable to our first layer embedding-based decision approach in terms of hallucination mitigation. However, our method requires only a single router decision during the forward pass, leading to a reduction in training time by approximately ∼30%. Furthermore, compared to Strategy III, our first layer-based decision mechanism more effectively reduces hallucinations. This improvement may be attributed to the fact that the first-layer representations process the input embeddings into activations that are more discriminative for hallucination detection.

## D.3 COMPARISON OF DIVERSE HALLUCINATION INDUCTION METHODS

We conduct a comparative experiment to validate the robustness of our method to different hallucination induction approaches. Specifically, we introduce hallucinations by providing the model with confusing instructions (Wang et al., 2024) during training. Table 10 shows that perturbing the instructions given to LVLMs can be effectively integrated with our approach, yielding comparable performance in hallucination mitigation. This indicates that our method is robust against different hallucination-inducing strategies.

## D.4 DATA SIZE FOR HIRE'S EDITOR TRAINING

To investigate the impact of training data size on model performance, we report hallucination mitigation results under varying training scales on the CHAIR benchmark. As shown in Fig. 6, the model's ability to mitigate hallucinations plateaus when the dataset size reaches about 2500 samples. As a result, performance cannot be further improved by large-scale training beyond this range. We argue that our approach requires no adjustments to the original LVLM parameters. Rather, it learns

Table 10: CHAIR evaluation of HIRE trained with different hallucination induction methods.

| Method | CHAIR$_S$ ↓ | CHAIR$_I$ ↓ |
|---|---|---|
| LLaVA-1.5 | 51.3 | 16.8 |
| Instruction perturbation | 32.4 | 8.7 |
| Visual perturbation | 30.2 | 9.7 |

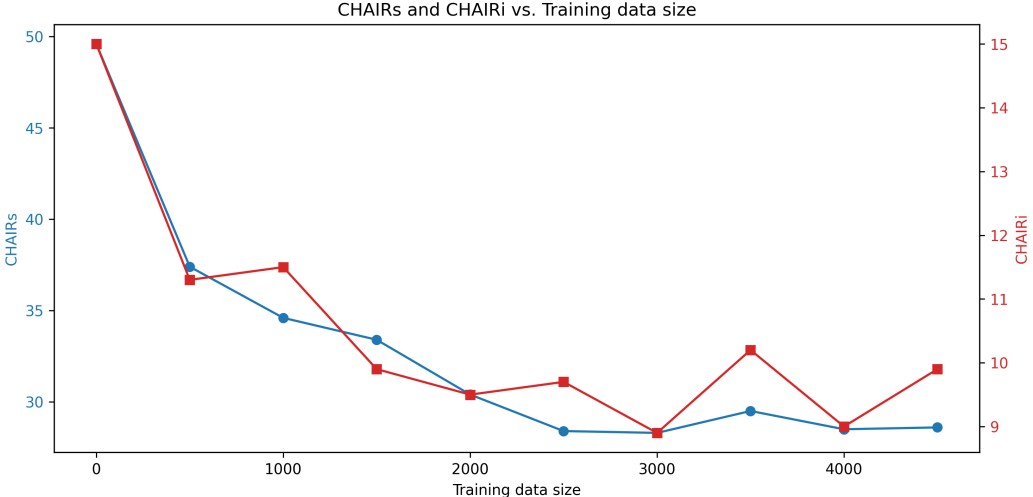

Figure 6: Performance of HIRE on CHAIR benchmark under different training data sizes.

an editing direction by training an editor and a router ( 0.05B parameters). Therefore, only a small number of samples is required to achieve effective hallucination mitigation with our method.

## D.5 HYPERPARAMETER SENSITIVITY ANALYSIS

To validate the training stability, we conduct additional experiments by retraining the model with different learning rates and scaling factors $\beta$, all without using a learning rate scheduler. As shown in Table 11, it can be found that our method shows low sensitivity to hyperparameters. Although certain settings can yield better performance, our primary focus is not to achieve SOTA results through hyperparameter tuning. Instead, we aim to provide the community with a novel perspective to rethink feature-level hallucination mitigation.

Table 11: Analysis of training stability under different hyperparameters.

| Setting | CHAIRs ↓ | CHAIRi ↓ |
|---|---|---|
| *Varying Learning Rate.* | | |
| LLaVA-1.5-7B | 51.3 | 16.8 |
| +Ours (1e-3) | 30.2 | 9.7 |
| 5e-4 | 32.4 | 12.1 |
| 5e-5 | 33.4 | 10.2 |
| *Varying Scaling Factor $\beta$.* | | |
| 0.05 | 31.6 | 10.1 |
| +Ours (0.1) | 30.2 | 9.7 |
| 0.15 | 32.4 | 10.3 |

Table 12: Evaluation on CHAIR and AMBER across model types and scales

| Model | CHAIR | | AMBER | | | |
|---|---|---|---|---|---|---|
| | CHAIR$_s$ ↓ | CHAIR$_i$ ↓ | CHAIR ↓ | Cover. ↑ | Cog. ↓ | HalRate ↓ |
| Qwen-2.5-VL | 46.2 | 9.6 | 8.8 | 59.9 | 3.4 | 46.6 |
| Qwen-2.5-VL+Ours | 35.6 | 8.1 | 6.5 | 60.3 | 2.1 | 36.8 |
| TinyLLaVA-1.5B | 58.6 | 17.5 | 11.2 | 50.8 | 6.2 | 48.0 |
| TinyLLaVA-1.5B+Ours | 36.6 | 11.7 | 8.6 | 48.7 | 2.6 | 35.6 |
| LLaVA-1.5-13B | 43.8 | 12.3 | 6.3 | 51.2 | 3.1 | 30.9 |
| LLaVA-1.5-13B+Ours | 29.8 | 8.2 | 4.6 | 49.0 | 1.7 | 22.0 |

Table 13: More results on the GQA and A-OKVQA datasets of POPE.

| Dataset | Method | Random | | Popular | | Adversarial | | All | |
|---|---|---|---|---|---|---|---|---|---|
| | | Acc↑ | F1↑ | Acc↑ | F1↑ | Acc↑ | F1↑ | Acc↑ | F1↑ |
| A-OKVQA | LLaVA-1.5 | 84.93 | 84.07 | 80.90 | 80.64 | 74.80 | 75.59 | 80.21 | 80.10 |
| | +VCD | 85.53 | 85.12 | 81.17 | 81.46 | 75.03 | 76.72 | 80.58 | 81.10 |
| | +M3ID | 85.06 | 84.30 | 80.90 | 80.77 | 74.80 | 76.15 | 80.25 | 80.41 |
| | +AVISC | 87.33 | 86.14 | 85.03 | 84.03 | **79.27** | 79.16 | 83.88 | 83.11 |
| | +Ours | **88.90** | **89.20** | **86.50** | **86.52** | 78.07 | **79.70** | **84.49** | **85.14** |
| GQA | LLaVA-1.5 | 84.80 | 84.16 | 79.37 | 79.64 | 76.00 | 76.89 | 80.08 | 80.23 |
| | +VCD | 85.63 | 85.38 | 78.73 | 79.78 | 76.40 | 78.15 | 80.25 | 81.10 |
| | +M3ID | 84.80 | 84.23 | 79.23 | 79.63 | 75.83 | 76.93 | 79.95 | 80.26 |
| | +AVISC | 87.40 | 86.21 | 83.33 | 82.54 | 80.37 | 80.00 | 83.70 | 82.92 |
| | +Ours | **88.87** | **89.21** | **84.67** | **85.21** | **80.63** | **82.07** | **84.72** | **85.50** |

## D.6 Evaluating Generalization Ability on Diverse Models and Datasets

To validate the generalization ability of our method, we present more experimental results on Qwen-2.5-VL-3B (Bai et al., 2025) and different scales of LLaVA series (Zhou et al., 2024; Liu et al., 2024b). It can be observed in Table 12 that our method still performs well across different types and scales of LVLMs.

We further present evaluation results on the GQA (Hudson & Manning, 2019) and A-OKVQA (Schwenk et al., 2022) datasets under the POPE benchmark in Table 13, comparing against previously reported results from Woo et al. (2024). Our method achieves consistent improvements in both accuracy and F1 score across these datasets.

We further conduct a cross-dataset experiment to evaluate the generalization ability of our method. Specifically, we randomly select 2,000 samples from Visual Genome (Krishna et al., 2017) (excluding overlaps with MSCOCO) to retrain our model and evaluate it on the MSCOCO dataset. Results in Table 14 show that our model maintains strong hallucination suppression performance even when trained on Visual Genome.

## D.7 Method Stability Analysis

To evaluate the stability of our method, we train the model five times with distinct random seeds and assess its performance on the CHAIR benchmark. As shown in Table 15, our approach consistently achieves stable results across different runs. On average, our method reduces CHAIR$_S$ and CHAIR$_I$ by 20.8 and 7.4, respectively, with variances as low as 1.3 and 0.31. These results demonstrate the high stability and reliability of our approach.

## D.8 Hallucination Controllability of Existing Methods

We focus on evaluating the hallucination control capability of post-hoc methods, specifically Contrastive Decoding (CD) and feature steering methods, as retraining-based methods lack this property. The evaluation is conducted on the CHAIR benchmark under a controlled setting (max new tokens = 512) using VCD Leng et al. (2024) and VTI Liu et al. (2025) as representative methods. The

Table 14: Evaluation on the generalization ability of HIRE

| Model | CHAIR$_S$ ↓ | CHAIR$_I$ ↓ |
|---|---|---|
| LLaVA-1.5 | 51.3 | 16.8 |
| +Ours (trained on VG_100K) | 30.6 | 10.5 |
| +Ours (trained on MSCOCO) | 30.2 | 9.7 |

Table 15: Evaluation on the stability of HIRE

| | CHAIR$_S$ ↓ | CHAIR$_I$ ↓ |
|---|---|---|
| baseline | 51.3 | 16.8 |
| 1 | 32.0 | 9.9 |
| 2 | 30.0 | 9.3 |
| 3 | 31.6 | 9.6 |
| 4 | 28.4 | 9.1 |
| 5 | 30.4 | 9.1 |
| Average | 30.5±1.3 | 9.4±0.31 |

results are shown as Fig 7 and Fig 8.It can be observed that the CD-based method exhibits unstable performance in hallucination control. While VTI demonstrates a certain level of hallucination control capability, its performance exhibits significant fluctuations as the parameter varies. To quantify the effect of hallucination control, we employ coefficients of determination $R^2$ (Draper, 1998) as an evaluation metric. Our method demonstrate superior hallucination control capabilities, with $R^2$ scores of 0.97 on CHAIRs and 0.96 on CHAIRi, compared to VTI's scores of 0.92 and 0.87, respectively.

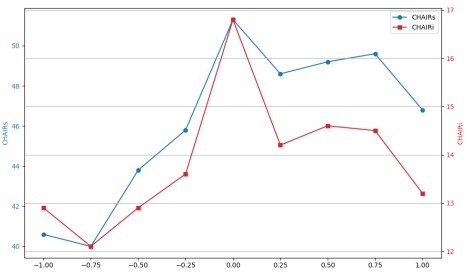

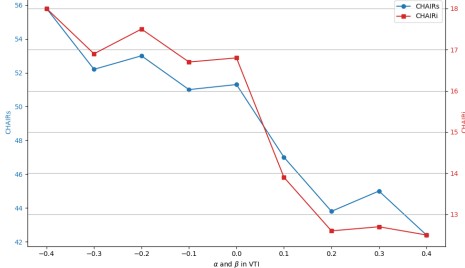

Figure 7: Control hallucination generation via the hyperparameter $\alpha$ in VCD (Leng et al., 2024).

Figure 8: Control hallucination generation via the hyperparameter $\alpha$ and $\beta$ in VTI (Liu et al., 2025).

## E   MORE QUALITATIVE ANALYSIS

To more explicitly demonstrate the effectiveness of our method in mitigating hallucinations in generative tasks, we visualize several examples, as illustrated in Fig. 9. For each image-prompt pair, we compare the responses generated by the original model and our method. Hallucinated words are highlighted in red, while newly identified objects that were missed by the original output are highlighted in blue to emphasize our method's enhanced perceptual accuracy. As shown in Fig. 10, we further show the ability of controlling hallucination of HIRE. Additionally, as shown in Fig. 11, we present several examples from discriminative tasks to further validate the robustness and generalization capability of our approach across different settings.

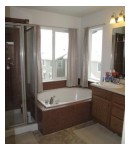

Q: Please describe this image in detail.

LLaVA-1.5: The image features a large, well-lit bathroom with a white bathtub and a shower stall. The bathtub is situated next to a window, allowing natural light to fill the room. The shower stall is located on the left side of the bathroom. There are several bottles placed around the bathroom, with some near the bathtub and others on the right side of the room. A toothbrush can be seen on the right side of the bathroom, and a cup is located near the center of the room. The bathroom appears to be well-maintained and clean.

Ours: The image features a large, clean bathroom with a white bathtub and a window. The bathroom is well-lit, with natural light coming in through the window. The tub is situated next to a sink, and there is a mirror above the sink. The bathroom also has a shower area, which is likely to be a separate room. The overall atmosphere of the bathroom is bright and inviting.

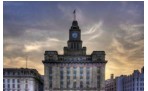

Q: Please describe this image in detail.

LLaVA-1.5: The image features a large, ornate building with a clock tower at the top. The clock is prominently displayed on the tower, making it a focal point of the building. The building is situated in a city, surrounded by other buildings and a street. There are several people in the scene, with some standing closer to the building and others further away. A car can be seen parked on the street, and a traffic light is visible in the background. The overall atmosphere of the image suggests a bustling urban environment.

Ours: The image features a large building with a clock tower prominently displayed on its side. The clock tower is situated near the top of the building, and it appears to be the focal point of the scene. The building has a unique design, and it is situated in a city with a sky background. The scene is set against a backdrop of a cloudy sky, giving the impression of a sunny day.

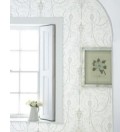

Q: Please describe this image in detail.

LLaVA-1.5: The image features a white room with a large window that allows natural light to fill the space. A vase filled with flowers is placed on a table near the window, adding a touch of color and life to the room. The room also has a doorway, which leads to another room. In addition to the vase, there is a potted plant in the room, providing a sense of nature and greenery. The combination of the flowers, the potted plant, and the natural light from the window creates a pleasant and inviting atmosphere.

Ours: The image features a white room with a large window that allows natural light to fill the space. The room is decorated with a beautiful floral wallpaper, and a vase with flowers is placed on the window sill. The vase is filled with flowers, adding a touch of color and elegance to the room. The combination of the white walls, the floral wallpaper, and the vase with flowers creates a serene and inviting atmosphere.

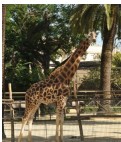

Q: Please describe this image in detail.

LLaVA-1.5: The image features a giraffe standing in a fenced enclosure, surrounded by trees. The giraffe is positioned in the center of the scene, with its long neck and legs visible. The enclosure appears to be a part of a zoo, as there are several other birds scattered throughout the area. Some birds are perched on the fence, while others are flying or walking around the enclosure. The scene captures the giraffe's natural habitat and the presence of other animals in the same space.

Ours: The image features a giraffe standing in a fenced-in area, possibly in a zoo. The giraffe is positioned in the center of the scene, with its long neck and legs prominently visible. The fenced area is surrounded by trees, providing a natural and serene environment for the giraffe. The giraffe appears to be looking at the camera, possibly enjoying the attention it receives.

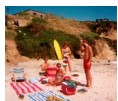

Q: Please describe this image in detail.

LLaVA-1.5: The image captures a group of people enjoying a day at the beach. There are four people in the scene, with two of them sitting on a blanket, and the other two standing nearby. They are all gathered around a surfboard, which is placed on the sand. The beach setting includes a chair positioned towards the left side of the image, and a backpack can be seen placed on the sand. The atmosphere appears to be relaxed and fun, as the group of friends spends time together on the beach.

Ours: The image depicts a group of people enjoying a day at the beach. There are four people in total, with two of them sitting on the sand, and the other two standing nearby. They are all wearing bathing suits and are relaxing on the beach. The beach scene also includes a surfboard, which is placed on the sand, and a chair, which is situated near the group of people. The overall atmosphere appears to be a fun and leisurely day spent at the beach.

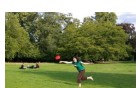

Q: Please describe this image in detail.

LLaVA-1.5: The image captures a woman in a green shirt playing with a red frisbee in a grassy field. She is in the process of throwing the frisbee, which is positioned in the middle of the scene. There are several other people in the background, some of them closer to the woman and others further away. A handbag can be seen placed on the ground near the left side of the scene. The overall atmosphere of the image is lively and fun, as everyone seems to be enjoying their time outdoors.

Ours: The image captures a woman in a green shirt playing with a red frisbee in a park. She is in the middle of throwing the frisbee, which is visible in the air. The park is surrounded by trees, providing a pleasant and natural setting for the activity. There are several other people in the park, some of whom are sitting on the grass, enjoying the outdoors. The scene is lively and full of energy, as people engage in various activities and enjoy the pleasant weather.

Figure 9: Some examples of generative tasks on the COCO dataset, with hallucinated content highlighted in red and newly added correct content displayed in blue.

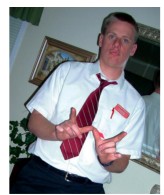

Q: Please describe this image in detail.

**Original output:** The image features a man wearing a white shirt and a red tie, standing in a room. He is making a hand gesture, possibly giving the middle finger, while looking at the camera. The man appears to be a waiter, as he is wearing a name tag. In the room, there is a potted plant located on the left side, and a clock can be seen hanging on the wall. The man is the main focus of the scene, and his hand gesture adds a sense of personality to the image.

**Positive edit:** The image features a man wearing a red tie and a white shirt, standing in front of a mirror. He is holding up his hand and making a peace sign with his fingers. The man appears to be enjoying himself and having a good time.

**Negative edit:** The image features a man wearing a white shirt and a red tie, standing in a room. He is making the "I Love You" hand gesture, with his fingers pointing upwards. The man is also wearing a name tag, which is visible on his shirt. In the room, there is a potted plant located on the left side, and a clock can be seen on the right side. A chair is also present in the scene, situated in the lower-left corner.

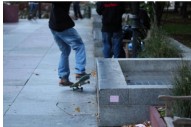

Q: Please describe this image in detail.

**Original:** The image captures a lively scene on a sidewalk where a young man is skillfully riding a skateboard. He is performing a trick, possibly a kickflip, as he moves down the sidewalk. There are several other people in the scene, some of them standing or walking nearby. A bench can be seen in the background, providing a place for people to sit and enjoy the atmosphere. Additionally, there are a few bottles scattered around the area, possibly belonging to the people present.

**Positive edit:** The image captures a young man skillfully riding a skateboard down a sidewalk. He is wearing a black jacket and appears to be enjoying his ride. There are several other people in the scene, some of whom are standing or walking nearby. The sidewalk is lined with benches, providing a comfortable space for people to sit and relax. The presence of multiple benches and people in the area suggests that this might be a popular spot for people to gather and socialize.

**Negative edit:** The image captures a man riding a skateboard down a sidewalk, with several people watching him perform the trick. The skateboarder is in the center of the scene, while the onlookers are scattered around him. There are a total of 11 people in the scene, with some standing closer to the skateboarder and others further away. A bench can be seen in the background, and a fire hydrant is also visible in the scene. In addition to the skateboarder, there is a person holding a cell phone, possibly capturing the moment or communicating with someone.

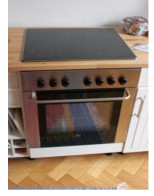

Q: Please describe this image in detail.

**Original output:** The image features a stove top oven with a black stove top and a glass window. The oven is located in a kitchen, and it appears to be empty. The stove top is surrounded by a wooden countertop. In the kitchen, there are a few items scattered around. A bottle can be seen on the left side of the oven, and a cup is placed on the right side. Additionally, there is a bowl located near the center of the scene.

**Positive edit:** The image features a stove top oven with a metal surface, placed on a wooden counter. The oven has a clear glass door, allowing us to see the interior. The stove is equipped with a set of knobs, which are likely to control the temperature and cooking functions. The oven is situated in a kitchen setting, and the countertop is made of wood.

**Negative edit:** The image features a stove with a glass door, which is open and revealing a blue and white towel. The stove is situated on a wooden countertop. The stove is equipped with a burner and a cooktop, and it is surrounded by a variety of utensils and appliances. There are several knives, with one located near the top left corner, another in the middle, and a third one on the right side. A spoon is also visible in the middle of the scene. A cup can be found on the left side of the stove, and a bowl is placed near the center. A toaster is present on the left side of the stove, and a book is located on the left side of the stove, possibly a cookbook or a recipe guide.

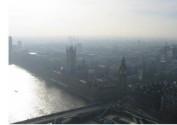

Q: Please describe this image in detail.

**Original:** The image showcases a beautiful cityscape with a large clock tower, Big Ben, towering over the city. The clock tower is situated in the middle of the scene, surrounded by other buildings. The city is bustling with activity, as numerous cars are scattered throughout the area, some closer to the foreground and others further in the background. In addition to the cars, there are two trucks visible in the scene, one near the center and the other towards the right side. The combination of the clock tower, the city's architecture, and the vehicles creates a lively and dynamic urban environment.

**Positive edit:** The image features a beautiful cityscape with a large clock tower prominently visible in the background. The clock tower is situated in the middle of the scene, surrounded by a variety of buildings. The cityscape is bustling with activity, as numerous cars are driving around the area, adding to the lively atmosphere. The scene also includes a bridge, which is a prominent feature in the background, and a river that runs through the city. The overall view of the cityscape is breathtaking, showcasing the city's architectural beauty and the dynamic nature of the urban environment.

**Negative edit:** The image showcases a bustling cityscape with a large clock tower, Big Ben, towering over the city. The clock is visible on the side of the tower, and the city is filled with various buildings, including a large castle. In the foreground, there is a busy street with cars and a truck, as well as a bus. A person can be seen walking on the sidewalk, and a traffic light is visible in the middle of the scene. The clock is positioned in the middle of the scene, and the cityscape is filled with buildings, creating a sense of depth and scale.

Figure 10: Some examples show that HIRE can amplify or mitigate hallucination by adjust the hyperparameter $\alpha$.

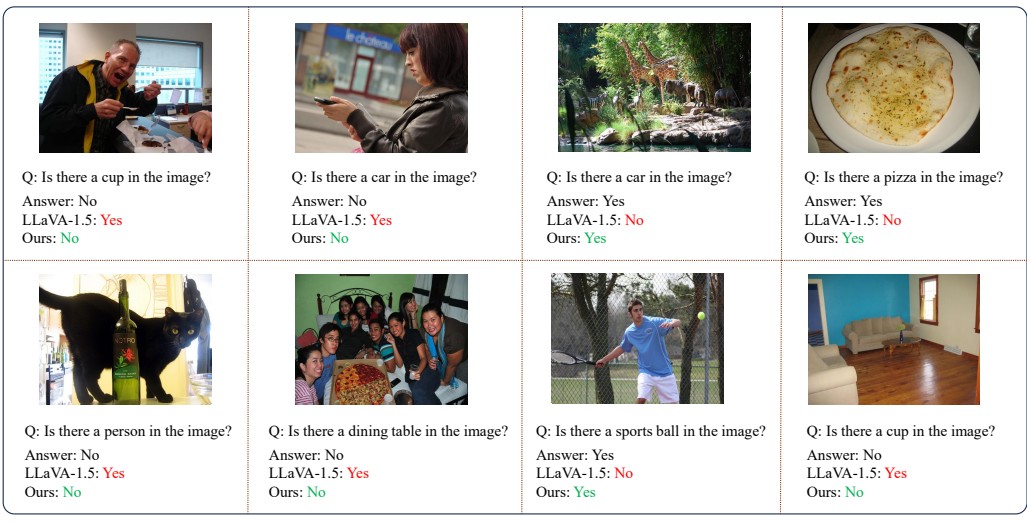

Figure 11: Some examples of discriminative tasks on the COCO dataset.

