# OpenReview forum: "Hallucination-aware Intermediate Representation Edit in Large Vision-Language Models"
_ICLR.cc/2026/Conference — ICLR 2026 Poster_

### Official Review · Reviewer_cpPf · 2025-10-26

**Soundness:** 3
**Presentation:** 3
**Contribution:** 3
**Rating:** 6
**Confidence:** 4

**Summary:**

This paper introduces a novel approach to mitigating hallucinations in language models by modifying intermediate representations, avoiding the need for retraining or doubling inference costs. The proposed framework, HIRE, includes 3 parts - Editor, Router, and Regulator, and dynamically detects and edits parts of the model’s internal states that are prone to hallucination, while allowing users to control the degree of editing to meet different requirements. The experiments demonstrate the effectiveness of HIRE, achieving state-of-the-art performance across three benchmarks.

**Strengths:**

* The paper presents convincing results, demonstrating the effectiveness of the proposed method across multiple benchmarks and settings.

* The writing is clear and well-structured, making the ideas and methodology easy to follow.

* The method is novel, introducing a new paradigm for hallucination mitigation by dynamically editing intermediate representations, which is both conceptually interesting and practically useful. The regulator is a nice addition that I haven't seen in other approaches.

* The analysis in Figure 5 effectively illustrates the key points, providing insight into how the editing of intermediate representations reduces hallucinations and supports the overall claims of the paper.

**Weaknesses:**

* Baseline comparison: As far as I understand, the baselines presented in the paper were trained on different datasets (or dataset sizes), making direct comparisons potentially misleading. A more appropriate comparison would include a reinforcement-learning baseline trained with LoRA on the same positive and negative examples that were used here, matching the number of trained parameters or FLOPs, to ensure a fair evaluation.

* Missing related work: The paper overlooks prior research on hallucination mitigation at the representation level (e.g., [1]). While most existing methods rely on retraining or contrastive decoding, some works directly edit representations and could serve as meaningful comparisons.

[1] Jiang et al., Interpreting and Editing Vision-Language Representations to Mitigate Hallucinations, ICLR 2025

**Questions:**

The model appears to scale well with increasing training data, as shown in Figure 6. This raises a couple of important questions:

* How does this scaling compare to other finetuning-based methods? Is it possible that the proposed approach is more data-efficient, extracting more value per training example?

* The curve does not appear to have saturated. At what data size would the model’s performance plateau, and what are the implications for larger-scale training?

---

> ### Author Response · Authors · 2025-11-21
>
> > Fairness of baseline comparison under matched training data and computational budget.
>
> Following your suggestion, we **retrain HIRE on HA-DPO [1] training set** to establish a fair comparison. The experimental results are presented below.
>
> | Method                             | CHAIRs $\downarrow$ | CHAIRi $\downarrow$ | Parameters |
> | ---------------------------------- | ------------------- | ------------------- | ---------- |
> | LLaVA-1.5-7B                       | 51.3                | 16.8                | 7.063B     |
> | + HA-DPO (LoRA)                    | 34.6 (-16.7)        | 10.1 (-6.7)         | 0.34B      |
> | + Ours (trained on HA-DPO dataset) | 30.4 (-20.9)        | 9.0 (-7.8)          | 0.048B     |
>
> Based on the table, our method achieves superior performance with fewer parameters than HA-DPO under the same training data.
>
>
>
> > Lack of comparison with representation-editing based hallucination mitigation methods.
>
> **Different Editing Focus**: Firstly, although both Projectaway[2] and our method aim to reduce hallucinations through feature editing, they operate on distinct modalities. Projectaway primarily focuses on editing image representations, whereas our approach specifically targets text representations. Furthermore, we incorporate a router to determine which tokens require editing, thereby improving inference efficiency. Finally, since the two methods address complementary aspects of the problem, HIRE **can be combined with Projectaway**. Experimental results of their integration are presented below.
>
> | Method               | CHAIRs $\downarrow$ | CHAIRi $\downarrow$ |
> | -------------------- | ------------------- | ------------------- |
> | LLaVA-1.5-7B         | 51.3                | 16.8                |
> | +Projectaway         | 43.8 (-7.5)         | 12.5 (-4.3)         |
> | + Ours               | 30.2 (-21.1)        | 9.7 (-7.1)          |
> | + Ours + Projectaway | 27.6 (-23.7)        | 8.3 (-8.5)          |
>
> It can be found that the combined strategy yields the strongest hallucination suppression. We will include the discussion in our final submission.
>
>
>
> > Regarding scaling comparison
>
> Conventional finetuning-based methods often require more than 10k samples (e.g., HA-DPO: 16k) to reach performance saturation.  While our method uses merely 2k samples, it matches or even surpasses the performance of existing finetuning approaches. We argue that finetuning-based methods require adjustments to all parameters of the LVLM, necessitating a large dataset for training. Conversely, our method bypasses this by training a compact editor and router (~0.05B parameters) to learn hallucination-reducing representations without modifying the original LVLM.
>
>
>
> > A more data-efficient, extracting more value per training example?
>
> We thank the reviewer for this insightful comment. We agree that selectively training on the most critical tokens and hidden states—a form of data-efficient training—represents a promising direction for further improving sample efficiency. While as a preliminary exploration, our focus was on establishing the feasibility of a new editing-based paradigm. We will explore dynamic training strategies that prioritize informative tokens/states as a central goal of our future research.
>
>
>
> > Clarification on performance plateau and data scalability
>
> Following your suggestion, we further expand the training set to include sizes ranging from 3,000 to 4,500 samples, with which we retrain the model. The relationship between the amount of training data and the degree of hallucination mitigation is presented in the table below.
>
> | Data_size    | CHAIRs $\downarrow$ | CHAIRi $\downarrow$ |
> | ------------ | ------------------- | ------------------- |
> | LLaVA-1.5-7B | 51.3                | 16.8                |
> | 2500         | 29.0 (-22.3)        | 9.9 (-6.9)          |
> | 3000         | 28.3 (-23.0)        | 8.9 (-7.9)          |
> | 3500         | 29.5 (-21.8)        | 10.2 (-6.6)         |
> | 4000         | 28.5 (-22.8)        | 9.0 (-7.8)          |
> | 4500         | 28.6 (-22.7)        | 9.9 (-6.9)          |
>
> The results indicate that the model's ability to mitigate hallucinations plateaus when the dataset size reaches 2500 to 4500 samples. As a result, performance cannot be further improved by large-scale training beyond this range. We argue that our approach requires no adjustments to the original LVLM parameters. Rather, it learns an editing direction by training an editor and a router (~0.05B parameters). Therefore, only a small number of samples is required to achieve effective hallucination mitigation with our method. We are happy to add these results to the supplementary materials if needed.
>
>
>
> [1] Beyond hallucinations: Enhancing lvlms through hallucination-aware direct preference optimization
>
> [2] Interpreting and Editing Vision-Language Representations to Mitigate Hallucinations. ICLR, 2025.

---

> > ### Comment · Reviewer_cpPf · 2025-11-27
> >
> > I thank the authors for their rebuttal. The rebuttal addressed my questions, and I will raise my score. I think that the new results should be added to the paper, and will make it much stronger.

---

### Official Review · Reviewer_z4GY · 2025-10-31

**Soundness:** 3
**Presentation:** 4
**Contribution:** 3
**Rating:** 6
**Confidence:** 3

**Summary:**

This paper introduces a new framework called HIRE for mitigating hallucinations in LVLMs without requiring model retraining or doubling inference costs. HIRE modifies intermediate representations by incorporating an Editor and a Router, which combine contrastive learning and DPO. By controlling the degree of hallucinations, the method adapts to different user requirements. Extensive evaluations show that the proposed approach achieves state-of-the-art performance on three benchmarks.

**Strengths:**

- The paper is well-written and easy to follow, with clear visualizations.
- The proposed HIRE framework is effective, lightweight, and applicable to most models.
- Detailed evaluations are conducted to demonstrate the effectiveness of HIRE.

**Weaknesses:**

- In Line 474, "..edited representations shift toward the non-hallucinated cluster and begin to merge, confirming that our editing effectively reduces hallucination." However, in Figure 5 (right), the entire green cluster appears to move closer to the separation line. How this observation can be explained should be clarified.
- The HIRE framework reduces hallucinations at the token level by enhancing/suppressing corresponding tokens based on the original/perturbed image. In the COCO dataset shown in Figure 9, the YES/NO tokens may not be as directly influenced as those in Figure 7, where the response directly contains instances. A deeper analysis on the effectiveness of HIRE in YES/NO questions could be constructed.

**Questions:**

- How does the centroid of the entire green cluster change relative to the separation line in Figure 5? Is it closer to the separation line after being edited?
- More analysis on the effectiveness of HIRE in YES/NO questions could be constructed.

---

> ### Author Response · Authors · 2025-11-21
>
> > W1&Q1: Green cluster closer to separation line
>
> Figure 5 illustrates the effect of our editing method. In the left panel, the red points (representing hallucinatory features) are mapped to the blue points (edited features) in the right panel. Spatially, the blue points are in close proximity to the green points (non-hallucinatory features, which remain fixed across both panels), demonstrating the effectiveness of our method in correcting the representations of hallucinations into truthful ones.
>
>
>
> > W2&Q2: Require for a deeper analysis on the YES/NO questions
>
> We respectfully clarify that our method is applicable to both generative and discriminative tasks.
>
> - **Generative tasks** involve prompts such as "Please describe this image in detail" that require LVLMs to produce long-form responses. HIRE decides and performs token-level editing sequentially throughout the generation process.
> - **For discriminative tasks**, which typically involve questions such as "Is there a {object} in the image?" requiring a direct "yes" or "no" answer, HIRE operates by editing only the hidden states of the last token in the input question. The LVLM then uses these edited representations to produce its final response.
>
> Therefore, Figure 9 provides a more concise presentation of the discriminative task instances.

---

### Official Review · Reviewer_AKoZ · 2025-10-31

**Soundness:** 3
**Presentation:** 2
**Contribution:** 2
**Rating:** 2
**Confidence:** 5

**Summary:**

The paper introduces a method to reduce object hallucination. Prior approaches, including training-based methods and contrastive decoding, often require substantial training resources or increase inference costs. To address these limitations, the authors propose HIRE, which is composed of an Editor and a Router. The Editor comprises two encoders, an attention module, and a decoder, and is used to compute a steering direction. The Router then determines whether to apply this steering direction. The proposed method is evaluated on three benchmark datasets: CHAIR, POPE, and AMBER, demonstrating its effectiveness in mitigating hallucinations.

**Strengths:**

- **Lower Inference Cost Compared to Contrastive Decoding.** Despite the increased number of parameters and computational overhead in HIRE, the overall inference cost remains lower than that of contrastive decoding methods, making it more efficient in practice.
- **Reasonable Design Choice.** The proposed router architecture is inspired by the Mixture of Experts (MoE) approach, a widely adopted and validated method in LVLMs and LLMs. By selectively activating Editor, the router enables adaptive and effective processing.

**Weaknesses:**

**W1. Lack of Comparison with Training Method.** The proposed method falls within the training approach. The detailed comparison with existing training approaches validate the effectiveness of the proposed method.
- How efficient is the proposed method from a training resource perspective compared to other training methods? Also, the performance comparison with training methods (e.g.,  HACL) is required than contrastive decoding method.
- Is M3ID equivalent to M3ID + DPO? If not, a direct comparison with M3ID + DPO should be provided.

**W2. Lack of Comparison with Steering Methods.** HIRE intervenes the latent representation by computing the direction of non-hallucination. The comparison with steering methods [R1, R2] is needed.

[R1] Le Yang et al., Nullu: Mitigating Object Hallucinations in Large Vision-Language Models via HalluSpace Projection, CVPR 2025

[R2] Sheng Liu et al., Reducing Hallucinations in Large Vision-Language Models via Latent Space Steering, ICLR 2025

**W3. Performance of General Task Capabilities.** LVLMs can perform various visual tasks in a zero-shot manner. After training, are these capabilities maintained?

**W4. Reproducibility.** HIRE has hyperparameters, including learning rates, schedulers, the scaling factor, and the editing strength. However, the paper does not provide a detailed justification for the selection of these hyperparameters. How were these values chosen?

**Questions:**

**Q1. Degree of Hallucination.** (Line 76) I think that most existing methods can control the degree of the hallucination. Both Contrastive Decoding (CD) and steering methods can effectively control the degree of hallucinations. Steering methods utilize a scaling factor for the steering direction, enabling control over the generated outputs. Similarly, CD has hyperparameters that influence the decoding process, allowing for the adjustment of the model’s logits.

**Q2. Missing Reference.** I was wondering whether the proposed method differs from MOEs; the paper does not cite existing work on MOEs.

---

> ### Author Response · Authors · 2025-11-21
>
> > W1: Lack of Detailed Comparison with Training Method
>
> In response, we highlight the following key differences in training resource requirements between our method and existing training approaches.
>
> 1. **Training Data Curation**: Existing training methods, such as HACL [1] and DPO-based approaches [2], require carefully curated pairs of positive and negative samples. This often involves using GPT-4 to generate negative samples. However, our method constructs negative samples by intervening in the model's inputs.
> 2. **Training Data Size**: Existing methods require large-scale training data (e.g., 558K for HACL pretraining, 16K for HA-DPO). In contrast, our approach achieves remarkable efficacy in hallucination reduction with only 2K samples.
> 3. **Computational Resource**: Current training methods, such as HACL, typically involve parameter updates to the LVLM, a process that demands substantial computational resources. In comparison, our approach trains an Editor and a Router while keeping the LVLM parameters frozen.
>
> **Performance Comparison**: We further present a comparison with HACL on both generative and discriminative task.
>
> #### MM-Hal
>
> | Method                   | Score $\uparrow$ | Hallucination rate $\downarrow$ |
> | ------------------------ | ---------------- | ------------------------------- |
> | LLaVA 1.5-7B (from HACL) | 2.08             | 0.52                            |
> | +HACL (from HACL)        | 2.13 (+0.05)     | 0.50 (-0.02)                    |
> | LLaVA 1.5-7B (reproduce) | 1.82             | 0.67                            |
> | +Ours                    | 2.03 (+0.21)     | 0.61 (-0.06)                    |
>
> #### POPE-all
>
> | Method                   | Acc $\uparrow$ | F1-score $\uparrow$ |
> | ------------------------ | -------------- | ------------------- |
> | LLaVA 1.5-7B (from HACL) | 87.19          | 86.16               |
> | + HACL (from HACL)       | 87.69 (+0.5)   | 87.26 (+1.1)        |
> | LLaVA 1.5-7B (from ours) | 82.04          | 80.42               |
> | + Ours (from ours)       | 87.27 (+5.23)  | 87.23 (+6.81)       |
>
> Due to differences in hyperparameters (such as input prompts and sampling methods), our baselines do not exactly match those of HACL. Therefore, we report both sets of results for a comprehensive comparison. It can be observed that our method demonstrates superior hallucination mitigation performance to HACL on both generative and discriminative tasks.
>
> **Comparison with M3ID+DPO**: The following compares our method with M3ID+DPO on the CHAIR benchmark (following M3ID, max new tokens set to 64).
>
> | Method            | CHAIRs $\downarrow$ | CHAIRi $\downarrow$ |
> | ----------------- | ------------------- | ------------------- |
> | LLaVA-7B          | 20.4                | 6.2                 |
> | LLaVA-7B M3ID+DPO | 13.5 (-6.9)         | 5.7 (-0.5)          |
> | LLaVA-7B +Ours    | 15.2 (-5.2)         | 5.4 (-0.8)          |
>
> As evidenced by the comparison, our method exhibits competitive performance. Moreover, it stems from a more streamlined process, avoiding the complex data curation and any updates to the LVLM's parameters.
>
>
>
> [1] Hallucination Augmented Contrastive Learning for Multimodal Large Language Model. CVPR, 2025.
>
> [2] Beyond hallucinations: Enhancing lvlms through hallucination-aware direct preference optimization

---

> ### Author Response · Authors · 2025-11-21
>
> > W2: Lack of Comparison with Steering Methods
>
> 1. **Comparison with Nullu**: Nullu [1] mitigates hallucinations by **adjusting model parameters**, whereas our approach operates by **editing intermediate activations**. Therefore, our method is **orthogonal** to Nullu and can be combined with it. The CHAIR evaluation results of the combined approach (following Nullu, max new tokens set to 64) are presented in the table below.
>
> | Method         | CHAIRs $\downarrow$ | CHAIRi $\downarrow$ |
> | -------------- | ------------------- | ------------------- |
> | LLaVA-1.5 7B   | 20.4                | 6.2                 |
> | + Nullu        | 17.0 (-3.4)         | 5.9 (-0.3)          |
> | + Ours         | 15.2 (-5.2)         | 5.4 (-0.8)          |
> | + Ours + Nullu | 13.2 (-7.2)         | 4.6 (-1.6)          |
>
> The combined approach achieves the best performance. This result demonstrates that the two methods can be integrated for a superposition of hallucination suppression.
>
> 2. **Comparison with VTI**:  While both our method and VTI [2] operate in the feature space, several key distinctions exist.
>    1. **Construction of Positive–Negative Pairs**: VTI relies on GPT model to generate hallucinated captions, which complicates the data preparation process. However, we adopt a simpler strategy of intervening at the model input to construct paired samples.
>    2. **Granularity of Editing**: VTI computes a single steering vector solely from the last token activation, assuming that all tokens share the same steering direction. In contrast, our method generates a dedicated editing vector for each token to ensure more precise and effective hallucination suppression.
>    3. **Adaptive Editing Strategy**: Recognizing that not all tokens are likely to induce hallucinations, we introduce a lightweight Router that avoids unnecessary edits on tokens with low hallucination risk, improving computational efficiency.
>    4. **Performance Evaluation**: We evaluate our method against VTI on both the CHAIR (following VTI, max new tokens set to 512) and POPE benchmarks.
>
> #### VTI comparison on CHAIR
>
> | Method       | CHAIRs $\downarrow$ | CHAIRi $\downarrow$ |
> | ------------ | ------------------- | ------------------- |
> | LLaVA-1.5-7B | 51.3                | 16.8                |
> | +VTI         | 35.8 (-15.5)        | 11.1 (-5.7)         |
> | +Ours        | 30.2 (-21.1)        | 9.7 (-7.1)          |
>
> #### VTI comparison on POPE
>
> | Method       | Acc $\uparrow$ | F1-score $\uparrow$ |
> | ------------ | -------------- | ------------------- |
> | LLaVA 1.5-7B | 82.04          | 80.42               |
> | +VTI         | 86.48 (+4.44)  | 85.87 (+5.45)       |
> | +Ours        | 87.27 (+5.23)  | 87.23 (7.81)        |
>
> Benefiting from our fine-grained editing and self-supervised design, the performance of our method consistently surpasses that of VTI on both generative and discriminative tasks.
>
>
>
> > W3: Performance of General Task Capabilities
>
> To verify whether the general capabilities are maintained, we evaluate the model on the MME (14 perception and cognition tasks) and SEED-Bench (12 tasks, e.g., spatial reasoning).
>
> | Method       | MME $\uparrow$ | SEED $\uparrow$ |
> | ------------ | -------------- | --------------- |
> | LLaVA-1.5 7B | 1751.64        | 64.3            |
> | +Ours        | 1751.99        | 63.8            |
>
> It can be observed that our method preserves the model's general capabilities on these challenging benchmarks.
>
>
>
> > W4: Reproducibility
>
> We want to clarify that Figure 3 in our paper includes an ablation study on edit strength. To further demonstrate the stability of our method to other hyperparameters, we retrain the model with three different learning rates (without a scheduler) and scaling factors.
>
> | learning_rate | CHAIRs $\downarrow$ | CHAIRi $\downarrow$ |
> | ------------- | ------------------- | ------------------- |
> | LLaVA-1.5-7B  | 51.3                | 16.8                |
> | +Ours (1e-3)  | 30.2 (-21.1)        | 9.7 (-7.1)          |
> | 5e-4          | 29.8 (-21.5)        | 9.5 (-7.3)          |
> | 1e-4          | 31.4 (-19.9)        | 10.3 (-6.5)         |
>
> | scaling_factor | CHAIRs $\downarrow$ | CHAIRi $\downarrow$ |
> | -------------- | ------------------- | ------------------- |
> | 0.05           | 31.6 (-19.7)        | 10.4 (-6.4)         |
> | +Ours (0.1)    | 30.2 (-21.1)        | 9.7 (-7.1)          |
> | 0.15           | 31.0 (-20.3)        | 9.9 (-6.9)          |
>
> It can be found that our method shows low sensitivity to hyperparameters. While certain settings can yield better performance, we would like to clarify that our primary focus is not to achieve SOTA results through hyperparameter tuning. Instead, we aim to provide the community with a novel perspective to rethink feature-level hallucination mitigation.
>
>
>
> [1] Nullu: Mitigating Object Hallucinations in Large Vision-Language Models via HalluSpace Projection, CVPR 2025.
>
> [2] Reducing Hallucinations in Large Vision-Language Models via Latent Space Steering, ICLR 2025.

---

> ### Author Response · Authors · 2025-11-21
>
> > Q1: Ability in Controlling Hallucinations
>
> To evaluate the hallucination control capability of the contrastive decoding (CD) and feature steering methods, we construct an experiment on the CHAIR benchmark (To ensure fair comparison, max new tokens set to 512) with VCD [1] and VTI [2]. The results for VCD are shown as follows.
>
> | cd_alpha (VCD)          | CHAIRs $\downarrow$ | CHAIRi $\downarrow$ |
> | ----------------------- | ------------------- | ------------------- |
> | -1.0                    | 40.6                | 12.9                |
> | -0.75                   | 40.0                | 12.1                |
> | -0.5                    | 43.8                | 12.9                |
> | -0.25                   | 45.8                | 13.6                |
> | 0.0                     | 51.3                | 16.8                |
> | 0.25                    | 48.6                | 14.2                |
> | 0.5                     | 49.2                | 14.6                |
> | 0.75                    | 49.6                | 14.5                |
> | 1.0 (default parameter) | 46.8                | 13.2                |
>
> It can be observed that the CD-based method exhibits unstable performance in hallucination control.
>
> Regarding steering methods, we compare their ability to control hallucinations relative to our method from both parameter-based and feature-based perspectives. Parameter-based methods, such as Nullu [3], often struggle to control hallucinations efficiently. As for feature-based methods, we conduct an experiment to evaluate their ability to induce hallucinations in a controlled manner. The results are as follows.
>
> **Controlling hallucination via $\alpha$ in VTI and HIRE**
>
> |          | VTI                 | VTI                 | HIRE                | HIRE                |
> | -------- | ------------------- | ------------------- | ------------------- | ------------------- |
> | $\alpha$ | CHAIRs $\downarrow$ | CHAIRi $\downarrow$ | CHAIRs $\downarrow$ | CHAIRi $\downarrow$ |
> | 0.4      | 42.4                | 12.5                | 44.8                | 13.7                |
> | 0.3      | 45.0                | 12.7                | 46.4                | 13.8                |
> | 0.2      | 43.8                | 12.6                | 48.4                | 14.4                |
> | 0.1      | 47.0                | 13.9                | 50.6                | 15.1                |
> | 0.0      | 51.3                | 16.8                | 51.3                | 16.8                |
> | -0.1     | 51.0                | 16.7                | 54.5                | 17.0                |
> | -0.2     | 53.0                | 17.5                | 58.0                | 17.4                |
> | -0.3     | 52.2                | 16.9                | 62.2                | 18.1                |
> | -0.4     | 55.8                | 18.0                | 63.8                | 18.5                |
>
> To quantify the effect of hallucination control, we employ coefficients of determination $R^2$ [4] as an evaluation metric. Our method demonstrate superior hallucination control capabilities, with $R^2$ scores of 0.97 on CHAIRs and 0.96 on CHAIRi, compared to VTI's scores of 0.92 and 0.87, respectively.
>
>
>
> > Q2: Lack of citing relevant MoEs paper
>
> We would like to clarify that although the module in our work shares the name "Router" with the one in Mixture-of-Experts (MoE), they operate on fundamentally different principles.
>
> 1. **Design Purpose**: Our design is motivated by the premise that not every token requires editing. Thus, our router identifies only high-risk tokens for editing. Meanwhile, routers in MoEs function to direct tokens to their most suitable expert network, thereby maintaining a clear separation of expertise.
> 2. **Optimization Strategy**：In conventional MoE models, the router is trained jointly with the main task objective, often accompanied by auxiliary constraints such as load balancing to encourage relatively uniform expert utilization. In contrast, the router in our approach is optimized using DPO, aiming to learn an optimal routing strategy that performs edits only when necessary.
>
> To prevent any potential confusion and following the reviewer's suggestion, we are open to renaming our module in the final submission to better highlight its unique role.
>
>
>
> We are happy to incorporate the above discussion into the final submission if needed.
>
>
>
> [1] Mitigating object hallucinations in large vision-language models through visual contrastive decoding. CVPR, 2024.
>
> [2] Reducing Hallucinations in Large Vision-Language Models via Latent Space Steering, ICLR 2025.
>
> [3] Nullu: Mitigating Object Hallucinations in Large Vision-Language Models via HalluSpace Projection, CVPR 2025.
>
> [4] Applied regression analysis.

---

> > ### Comment · Reviewer_AKoZ · 2025-11-28
> >
> > I acknowledge the authors' rebuttal and most of my concerns have been addressed. I will adjust my score. Since the rebuttal policy allows for uploading the revised PDF of the paper, could authors please incorporate the new experiments into the main paper and upload the revised PDF?

---

### Official Review · Reviewer_GT68 · 2025-11-01

**Soundness:** 3
**Presentation:** 3
**Contribution:** 3
**Rating:** 6
**Confidence:** 4

**Summary:**

The paper proposes HIRE, a lightweight plug-in module for large vision-language models that reduces hallucinations without retraining the base model or doing multi-pass decoding. Instead of modifying output logits, HIRE edits the model’s intermediate hidden representations during inference. It learns to disentangle each token’s “semantic content” from its “hallucination tendency,” computes an edit direction that shifts the token’s representation toward image-grounded truth, and applies this edit only when a learned router predicts the token is likely hallucinated. A single global coefficient controls the edit strength and direction, enabling continuous, user-adjustable hallucination suppression. Experiments on standard hallucination benchmarks (e.g., CHAIR, POPE, AMBER) show that HIRE significantly lowers hallucination rates while preserving fluency and with minimal compute overhead.

**Strengths:**

1. This work performs hallucination control by directly editing intermediate representations, rather than retraining the whole LVLM or running contrastive decoding with multiple forward passes. This keeps compute cost low.
2. Authors disentangle “semantic content” vs. “hallucination tendency” in the hidden space and edits only the hallucination component, which preserves fluency and factual content instead of bluntly suppressing all tokens.
3. This work improves standard hallucination benchmarks (CHAIR, POPE, AMBER) while keeping language natural, showing that hallucination mitigation does not have to trade off descriptiveness.

**Weaknesses:**

1. Training images come from MSCOCO, and some evaluation benchmarks (e.g., POPE) also draw from MSCOCO. The method should be tested on images from other datasets to rule out dataset bias and show robustness.
2. The approach is only demonstrated on older LVLM backbones （LLaVA 1.5 & InstructBLIP. It should also be tested on the latest multimodal LLMs (e.g., the Qwen 2.5 VL series) to further validate generality and effectiveness.
3. The paper does not report the number of additional training parameters introduced by the added modules.

**Questions:**

Please refer to the weaknesses part.

---

> ### Author Response · Authors · 2025-11-21
>
> > W1: Concerns on method robustness
>
> To evaluate the robustness of our method, we present additional POPE experiments on the A-OKVQA and GQA datasets (sourced from Visual Genome) **in Appendix D.5 and Table 9**. Additionally, we randomly selected 2,000 samples from Visual Genome to retrain our model and evaluate it on the MSCOCO dataset, as shown **in Appendix D.5 and Table 10**. The results demonstrate that our method continues to perform well in reducing hallucinations, confirming its strong robustness.
>
>
>
> > W2: More experiments on new models
>
> Additional experiments validating our method's effectiveness on new models are presented **in Table 8 and Appendix D.5**, showcasing its performance on the Qwen2.5-VL model across the CHAIR and AMBER generative tasks. The results show that our approach effectively mitigates hallucinations even when applied to this new architecture, demonstrating its consistent performance across different models.
>
>
>
> > W3: Lack of reporting the number of additional training parameters
>
> In response to your comment, we provide the parameter count of LLaVA-1.5-7B and the total number of trainable parameters required by our method.
>
> | Method       | Parameters     |
> | ------------ | -------------- |
> | LLaVA-1.5-7B | 7.063B         |
> | HIRE         | 0.048B (~0.7%) |
>
> The results show that our approach involves training only 0.048B parameters. This amounts to a ~0.7% of the total parameters in LLaVA-1.5-7B. If you consider it necessary to report the number of additional training parameters, we will add this to our final submission.

---

### Meta-Review · Area_Chair_h4Ck · 2025-12-18

**Summary:**

This paper introduces HIRE, a novel, lightweight plug-in framework designed to reduce hallucinations in Large Vision-Language Models (LVLMs) during inference. Unlike previous methods that require model retraining or multi-pass decoding, HIRE modifies intermediate representations by selectively editing tokens predicted to be hallucinated. It achieves this by separating a token's semantic content from its hallucination tendency and applying edits to only those tokens deemed to be problematic. This approach is controlled by a global coefficient that allows users to adjust hallucination suppression. The effectiveness of HIRE is demonstrated through experiments on three standard benchmarks (CHAIR, POPE, AMBER), showing significant reductions in hallucination rates while preserving fluency, with minimal computational overhead. The authors have addressed all significant concerns raised by the reviewers, providing additional experiments, clarifications, and comparisons. Therefore, I recommend accepting this paper for publication.

**Reviewer Concerns:**

Almost all reviewer concerns were addressed, summarized as follows:

• Reviewer GT68 (wei chen):

Core Concern: The method was primarily tested on MSCOCO and POPE datasets, raising concerns about dataset bias. The reviewer also pointed out that the method was only evaluated on older LVLMs (LLaVA 1.5, InstructBLIP), and suggested testing on newer models like Qwen 2.5 VL. Additionally, the number of additional parameters introduced by the method was not reported.

Authors' Response Summary: The authors addressed the dataset bias concern by testing on A-OKVQA and GQA datasets, showing robust performance. They also evaluated Qwen 2.5 VL, confirming HIRE's effectiveness on newer models. The number of additional parameters (0.048B) was provided, clarifying the minimal overhead.

• Reviewer AKoZ (Nam Hyeon Woo):

Core Concern: The reviewer sought a more detailed comparison with training-based methods like HACL and DPO, as well as with steering methods like Nullu and VTI. There was also concern about whether the method maintained general task capabilities post-training and a lack of justification for the choice of hyperparameters.

Authors' Response Summary: The authors provided a comparison with HACL and M3ID+DPO, showing superior performance with fewer parameters. They also demonstrated that HIRE preserves the general capabilities of LVLMs on additional tasks and clarified the robustness of their method to different hyperparameters, providing an ablation study.

• Reviewer z4GY (Jia Hongrui):

Core Concern: The reviewer questioned the interpretation of the cluster shift in Figure 5 and asked for a deeper analysis of HIRE's performance on YES/NO questions, as these may not be directly impacted by token-level edits.

Authors' Response Summary: The authors clarified that the shift in the green cluster represented the correction of hallucinated features to more truthful ones. They also explained that for YES/NO questions, HIRE edits only the last token’s hidden state, ensuring the accuracy of the responses.

• Reviewer cpPf (Yossi Gandelsman):

Core Concern: The reviewer raised issues with baseline comparisons, suggesting that the methods should be tested on the same dataset with matching computational resources. Additionally, the reviewer noted a lack of comparison with representation-level hallucination mitigation methods and inquired about HIRE's data efficiency and performance plateau.

Authors' Response Summary:The authors retrained HIRE on the HA-DPO dataset, showing superior performance with fewer parameters and samples. They also compared HIRE with Projectaway, highlighting their complementary nature and combining them for improved hallucination suppression. Finally, they demonstrated that HIRE performs well with only 2k samples, and its performance plateaus around 2,500 to 4,500 samples.

**Reviewer Scores:**

I believe that if each reviewer had actively participated in the discussion, they would likely have made positive changes to their scores. Below is a brief analysis of why each reviewer would likely change their score:

Reviewer GT68 (wei chen): The authors conducted additional experiments on new datasets (A-OKVQA, GQA) and newer models (Qwen 2.5 VL), confirming the robustness of the method across different datasets and models. This additional validation likely addressed the reviewer’s concern about dataset bias and generalizability.

Reviewer AKoZ (Nam Hyeon Woo): The authors provided clear comparisons with training-based methods (HACL) and M3ID+DPO, as well as steering methods like Nullu and VTI, addressing the reviewer’s concern about method comparison. These comprehensive comparisons likely would lead to a positive adjustment in the score.

Reviewer z4GY (Jia Hongrui):  The authors clarified the interpretation of the cluster shift in Figure 5 and explained how HIRE performs on YES/NO questions, addressing the reviewer’s request for more analysis. This clarity likely would lead to a positive change in score.

Reviewer cpPf (Yossi Gandelsman): The authors provided a fair comparison with HA-DPO and demonstrated that HIRE performs well with fewer parameters. They also showed that HIRE can be integrated with Projectaway to improve hallucination suppression, addressing concerns about baseline fairness and data efficiency. These additions likely would result in a higher score.

---

### Decision · Program_Chairs · 2026-01-26

Accept (Poster)